# A machine-vision approach for automated pain measurement at millisecond timescales

**Jessica M Jones[1†], William Foster[1†], Colin R Twomey[1†], Justin Burdge[1], Osama M Ahmed[2], Talmo D Pereira[2], Jessica A Wojick[3], Gregory Corder[3], Joshua B Plotkin[1], Ishmail Abdus-Saboor[1]\***

[1]Department of Biology, University of Pennsylvania, Philadelphia, United States; [2]Princeton Neuroscience Institute, Princeton University, Princeton, United States; [3]Departments of Psychiatry and Neuroscience, University of Pennsylvania, Philadelphia, United States

**Abstract** Objective and automatic measurement of pain in mice remains a barrier for discovery in neuroscience. Here, we capture paw kinematics during pain behavior in mice with high-speed videography and automated paw tracking with machine and deep learning approaches. Our statistical software platform, PAWS (Pain Assessment at Withdrawal Speeds), uses a univariate projection of paw position over time to automatically quantify seven behavioral features that are combined into a single, univariate pain score. Automated paw tracking combined with PAWS reveals a behaviorally divergent mouse strain that displays hypersensitivity to mechanical stimuli. To demonstrate the efficacy of PAWS for detecting spinally versus centrally mediated behavioral responses, we chemogenetically activated nociceptive neurons in the amygdala, which further separated the pain-related behavioral features and the resulting pain score. Taken together, this automated pain quantification approach will increase objectivity in collecting rigorous behavioral data, and it is compatible with other neural circuit dissection tools for determining the mouse pain state.

**\*For correspondence:**
ishmail@sas.upenn.edu

[†]These authors contributed equally to this work

**Competing interests:** The authors declare that no competing interests exist.

## Introduction

Numerous genetic and environmental factors shape the subjective experience of pain. While humans can articulate the intensity and unpleasantness of their perceived pain in the form of pain scales and questionnaires (*Attal et al., 2018*; *Melzack, 1987*), determining pain states in non-verbal animals remains a significant challenge. Rodents are the predominant model organism to study pain and there is an urgent need to develop high-throughput approaches that accurately measure pain. The past 50 years of pain research have relied on the paw withdrawal reflex metric to measure pain-related behaviors in rodents, which has contributed to important discoveries about nociception (*Basbaum et al., 2009*; *Deuis et al., 2017*). However, the traditional approach of manually scoring paw lifting suffers from an inability to determine whether paw movement away from a stimulus is motivated by the experience of pain. Improving the resolution, and increasing the dimensionality, of the common paw withdrawal assay has the potential to increase the predictive validity of translational pain therapeutics and to increase the rate at which basic science findings are translated to the clinic.

Animals generate rapid motor responses to somatosensory stimuli at millisecond speeds that cannot be readily detected by eye (*Severson et al., 2017*; *Douglass et al., 2008*). Therefore, significantly increasing the recording rate of the motor actions, coupled with the sub-second mapping of behavioral signatures, will sharpen the resolution and confidence for assessing an animal's internal

pain state. For example, researchers recorded optogenetically-induced nociceptive behaviors at 240 frames per second (fps), which facilitated the precise mapping of nocifensive behaviors including paw withdrawal, paw guarding, jumping, and vocalization (*Arcourt et al., 2017*). Two additional studies recording between 500 and 1000 fps using both natural and optogenetic nociceptive stimuli, demonstrated that nociceptive withdrawal latencies were on the order of 20–130 milliseconds (ms) (*Blivis et al., 2017*; *Browne et al., 2017*). More recently, we recorded mouse somatosensory behaviors at 500–1000 fps, coupled with manual behavioral mapping, statistical modeling, and machine learning to create a more objective 'pain scale' (*Abdus-Saboor et al., 2019*). Although these studies provide a framework for using high-speed videography for fine-assessment of pain, a major limitation lies in the relatively low-throughput nature of manual scoring of the video frames, which adds potential human error, and limits the ease of platform adoption in other laboratories.

Recently, computational neuroethology platforms have introduced a suite of machine learning and deep neural networks to automatically track animal body parts during behavior for postural estimation (*Datta et al., 2019*). Platforms such as Motion Sequencing (MoSeq) use three-dimensional depth imaging, quantitative analyses, and fitting with unsupervised computational models to estimate animal posture within an open arena and can automatically reveal ~60 unique sub-second behavioral signatures (*Wiltschko et al., 2015*). DeepLabCut and LEAP, train deep neural networks (DNN) with relatively limited training datasets, allowing the computer to accurately track unlabeled body parts such as a mouse paw, ear, or even a single digit through many frames of videography data (*Pereira et al., 2019*; *Mathis et al., 2018*). Alternatively, the markerless automated tracking software ProAnalyst tracks moving objects across high frame rate videography data (*Tiriac et al., 2012*; *Libby et al., 2012*). This approach does not use deep learning but relies on built-in machine learning algorithms for automated tracking, which provides an easier point of entry for researchers with limited time for software development or computing power.

Here, we present an automated mouse pain scale that combines videography at 2000 fps, automated paw tracking with ProAnalyst and SLEAP, and new software called PAWS (Pain Assessment at Withdrawal Speeds), which automatically scores seven defined behavioral features and produces a resulting univariate pain score. Beginning with seven commonly used genetically inbred mouse strains we revealed stereotyped sub-second paw trajectory patterns, with simple up-down lifts typifying the response to innocuous stimuli and elaborate sinusoidal patterns typifying the responses to noxious stimuli. By projecting paw position onto the time-varying principal axis of paw movement, we identified shaking behavior as simple sequences of peaks and valleys in this univariate time series, and paw guarding as extended periods of stasis devoid of shaking before returning the paw to the ground.

After building an automated pain assessment platform, we confirmed that the seven movement features we automatically measured were sufficient to separate behavioral responses to innocuous touch from noxious pinprick stimuli. Moreover, this could be accomplished using a single univariate measure of pain identified as a linear transformation of the seven behavioral features, by ordinal logistic regression. This same univariate scale further segregated noxious intensity. Cross-validation of our univariate pain scale confirmed that PAWS performs well for decoding the stimulus type and intensity across mouse genetic lines based on the animal's sub-second behavioral responses. Finally, using our recently described protocol to gain genetic access to basolateral amygdala (BLA) neurons that are responsive to pain (*Corder et al., 2019*), we chemogenetically activated the BLA pain ensemble to validate our platform's ability to detect centrally driven pain responses. Thus, we can automatically measure increases in mechanical pain responsiveness to noxious stimuli while manipulating central pain circuits. Taken together, this work reveals that automating paw tracking and subsequent quantification of pain behaviors with high frame rate videography provides a reliable method to objectively determine the mouse pain state.

## Results

### High-speed videography and automated paw tracking during evoked behaviors

To capture sub-second behavioral ethograms during somatosensory behaviors in freely behaving mice, we recorded mice at 2000 fps, with a particular focus on the stimulated paw. We reasoned

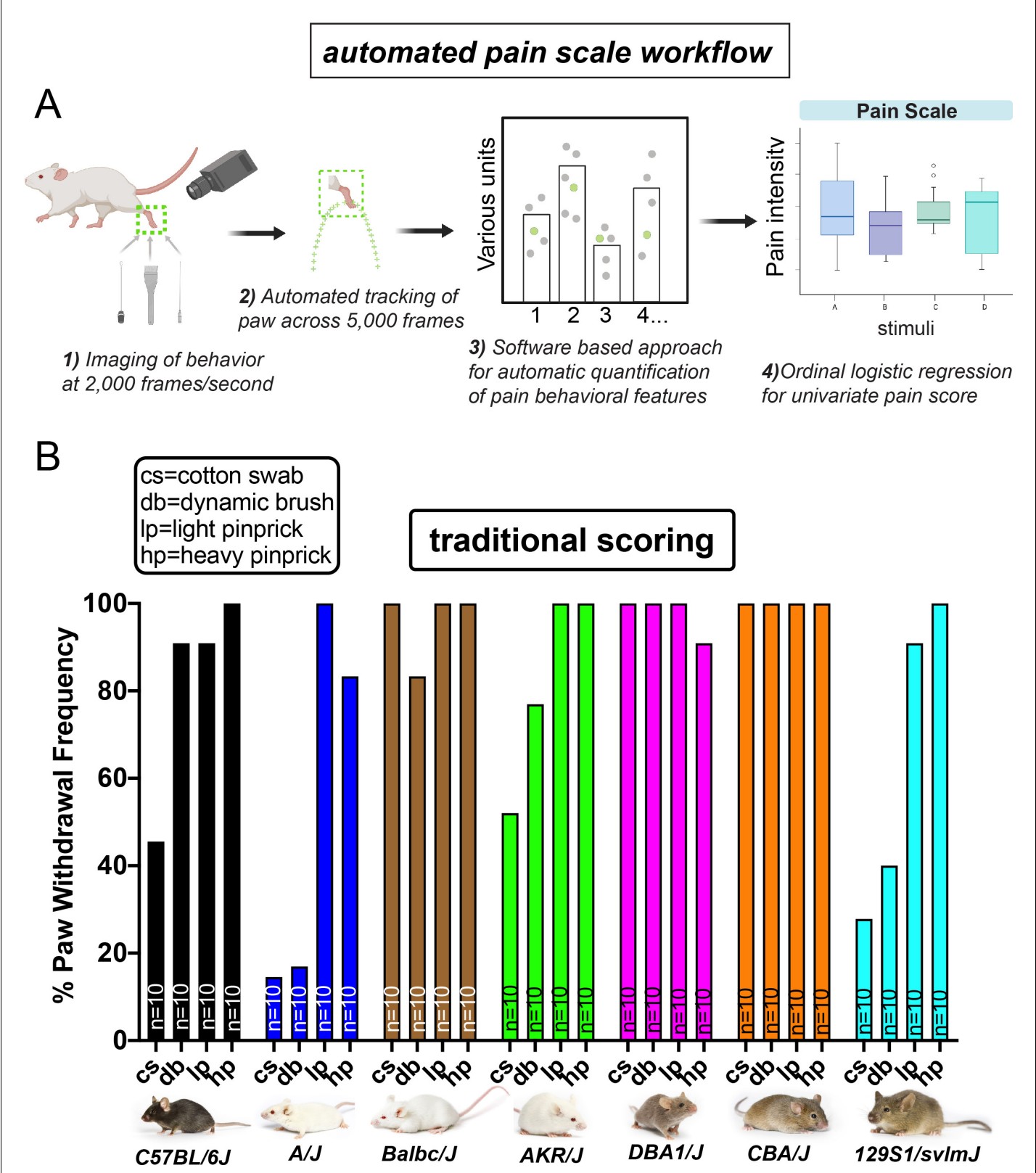

**Figure 1.** Automated pain assessment workflow in comparison to traditional unidimensional pain scoring. (**A**) Workflow pipeline in following order consisting of 1) high-speed videography of freely behaving mice, 2) machine/deep learning-based approaches for automatic tracking of the stimulated paw, 3) PAWS software for automatic quantification of defined pain behavioral features, 4) statistical modeling with ordinal logistic regression for

*Figure 1 continued on next page*

Figure 1 continued
separation of touch versus pain on a trial-by-trial basis. (B) Traditional scoring focused on paw withdrawal frequencies to four mechanical stimuli: cs = cotton swab, db = dynamic brush, lp = light pinprick, hp = heavy pinprick. N = 10 mice per strain. Images from Jackson laboratories.

that we could develop a pipeline where we first performed behavior experiments, followed by automated paw tracking and automated pain scoring, and lastly statistical modeling to transform multidimensional datasets into a single dimension that separated touch from pain (*Figure 1A*). To begin, we used 10 mice from seven commonly used inbred lines and on separate days to avoid sensitization to the stimuli, we applied to one hind paw a static innocuous stimulus (cotton swab), a moving innocuous stimulus (dynamic brush), a weak noxious stimulus (light application of a pinprick), and an intense noxious stimulus (heavy application of a pinprick). We used pinprick instead of von Frey hair filaments (VFHs) based on our prior studies showing that pinprick stimuli activate nociceptors and consistently evoke painful responses, whereas VFHs were more variable depending upon the force applied (*Abdus-Saboor et al., 2019*). Using traditional pain scoring, we noticed that the paw withdrawal frequencies to these four stimuli varied widely across these seven strains. For example, Balb/cJ, DBA1/J, and CBA/J displayed high paw withdrawal rates to all mechanical stimuli, both the innocuous and the two noxious (*Figure 1B*). Conversely, C57BL/6J and AKR/J displayed high paw withdrawal rates to both the noxious stimuli and the innocuous dynamic brush, but not to a cotton swab (*Figure 1B*). Finally, A/J and 129S1 mice displayed high rates of paw lifting to both pinprick stimuli and low withdrawal rates to the two touch stimuli (*Figure 1B*). Since mice will move their paw to both innocuous and noxious stimuli, it is hard to determine if these differences in withdrawal frequencies are driven by genetic differences in susceptibility to pain. What these data likely reveal using the common traditional approach is that this test in isolation may be an inadequate measurement of pain at baseline states.

Next, we turned to automated tracking with the seven mouse strains to determine the X, Y coordinates of the paw across approximately 5000 frames – recording at 2000 fps with total behavior time from stimulus application to paw lift and return to the floor being approximately 2–3 s. Using machine learning algorithms embedded within the ProAnalyst motion tracking software, we manually labeled the center of the stimulated paw in each video and the machine automatically tracked the paw throughout each additional frame (*Figure 2A–B'*) (see *Figure 2—video 1*). While observing the automated paw trajectory patterns, we noticed that stereotyped motor sequences defined the movement away from the four stimuli, regardless of strain background (*Figure 2A–B'*). For example, the responses to the two innocuous stimuli were typically up-down C-shaped movements (*Figure 2A–B'*). Conversely, the responses to the two noxious stimuli were typically more elaborate movements, often accompanied by an orbital tightening of the eye, which is a known facial feature of intense pain (*Figure 2A–B'; Langford et al., 2010*). Only the 129S1 strain showed responses to pinprick that were devoid of sinusoidal irregular paw trajectory patterns (*Figure 2Y–B'*). In the majority of tested strains, we also noticed that the paw trajectory pattern in response to a weakly painful stimulus (light pinprick) often resulted in a figure-eight like sequence that may point toward a shared sensorimotor neural circuit that governs how animals respond to weak and painful stimuli given their body posture and space constraints (*Figure 2A–B'*).

Next, we randomly chose one strain (AKR/J) and we used a deep learning-based pose tracking algorithm called SLEAP (in preparation, based on *Pereira et al., 2019*), to predict mouse toe and mid-paw positions during somatosensory behaviors recorded at high speed (*Figure 2C'-F'*). Our mouse paw-tracking model was generated from a small training set of video frames collected from the four assays (~9.5% of video frames per assay). In general, the paw trajectory patterns with SLEAP resemble those of ProAnalyst, and the software package we describe below is compatible with automated tracking data from either tool or even others. Taken together, our ability to detect clear qualitative distinctions in paw movements with automated tracking approaches gave us confidence that we could use spatiotemporal data of paw position to automatically extract features that may be useful in determining the mouse pain state. Additionally, these data demonstrate that the nature of the response when an animal withdrawals its paw might be a more reliable indicator of pain state than the number of times the animal lifts its paws to a given stimulus. In other words, paw withdrawal frequencies revealed great variation across the seven strains (*Figure 1*), whereas paw trajectory patterns to a given stimulus showed quite stereotyped responses among these strains (*Figure 2*).

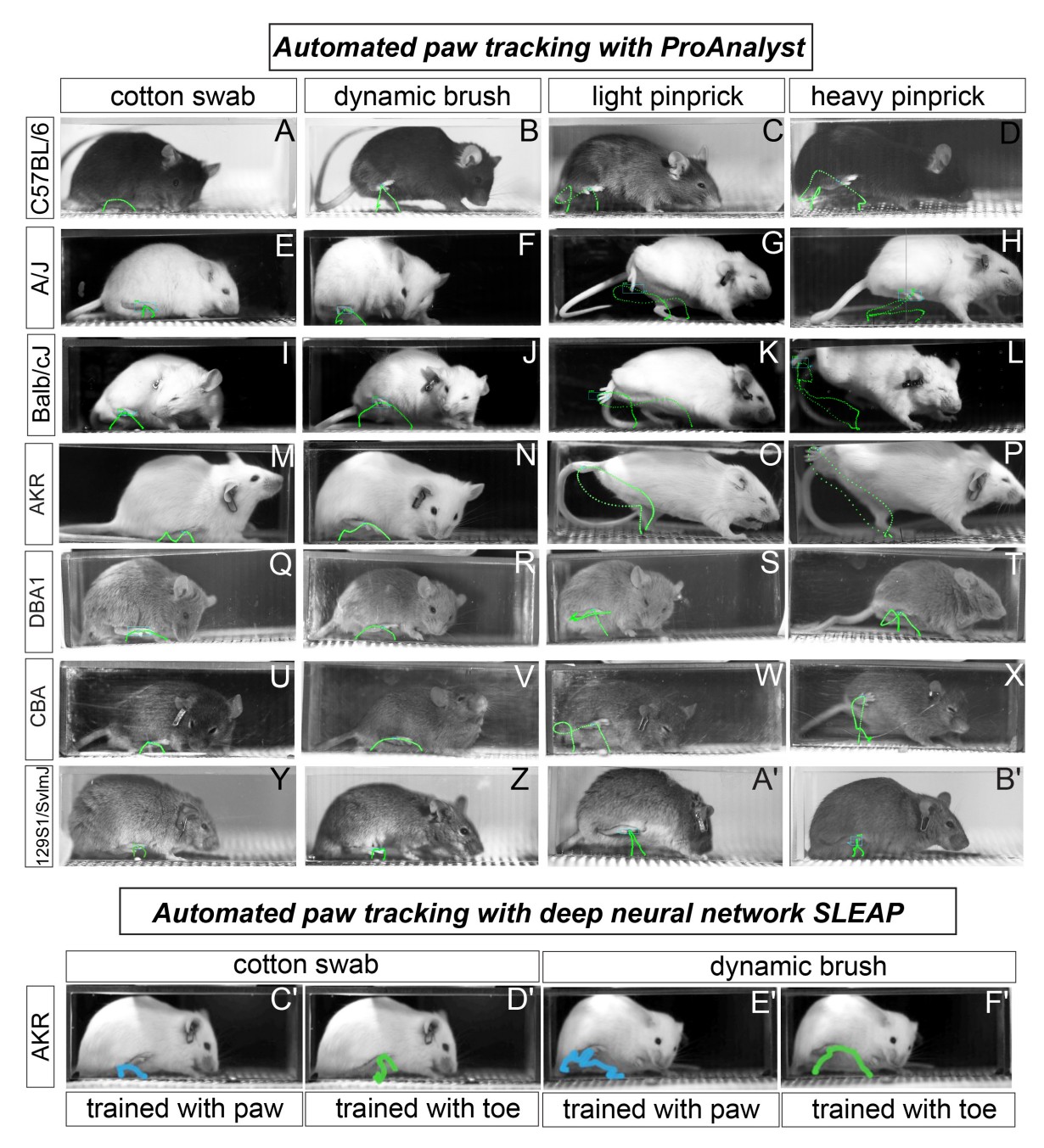

**Figure 2.** Automated paw tracking with a high-speed recording of behavior. (**a-b'**) ProAnalyst machine learning-based paw tracking. (**y-z'**), SLEAP deep neural network based paw or toe tracking. All still images represent a single frame of ~5000 total frames. Green and blue lines display paw trajectory patterns during the entire behavior. N = 10/mice per stimulus and images shown are representative of each strain.

The online version of this article includes the following video for figure 2:

**Figure 2—video 1.** Automatic paw tracking with the delivery of noxious mechanical stimulus.
https://elifesciences.org/articles/57258#fig2video1

## Development of software to automatically score pain behavioral features

We developed software to systematically quantify the seven pain-relevant features of the paw position time series based on existing measurements in the literature (*Abdus-Saboor et al., 2019*).

Maximum paw height, lateral velocity, vertical velocity, and the total distance traveled by the paw, were all computed based on a polynomial smoothing (Savitsky-Golay filter of order 3) of the original paw position time series. These four features were computed for two different windows of the paw trajectory time series: the time leading up to the initial peak in paw height (time $t^*$ in *Figure 3A*), and the time following the initial paw peak, which we refer to as pre-peak and post-peak, respectively. The $t^*$ designation reflects a biological designation: behavior before $t^*$ corresponds to a reflexive rapid withdrawal of the paw, whereas behavior following $t^*$ includes lingering paw attending behaviors that require some degree of conscious perception. In the post-peak time window, we also identified periods of 'shaking' and 'guarding' based on a threshold displacement along the principal axis of the paw movement (*Figure 3B and C*). We used this delineation to quantify the total duration spent shaking or guarding as well as the total number of paw shakes across shaking periods. We refer to this software package as PAWS).

## Automated scoring of rapid paw dynamics and lingering pain behaviors

With software generated to automatically measure paw movement features related to the mouse pain state, we plotted and analyzed the data across the seven mouse strains with the four mechanical stimuli described above. For plotting the individual behavioral features, we separated the paw distance traveled measurement into pre-peak and post-peak distances. To standardize across varying units and to appreciate the individual deviation from the median, we transformed the raw output measurements into Z-scores (*Figure 4*). The first readily apparent feature we noticed was that all measurements across these seven strains were interspersed without clear separation among strains, despite the vast differences in paw withdrawal frequencies to these same stimuli (*Figure 1B*). These data suggest that these seven strains are phenotypically quite similar in their acute responses to mechanical stimuli despite the genetic variation that distinguishes these mouse lines. In regards to pre-paw peak behavioral features, we observed a statistical separation in the stimulated paw's velocity on both the X and Y axes between touch (CS, DB) and pain stimuli (LP, HP) as the paw withdrew upwards to reach its first highest peak (*Figure 4A–C*). The mean max height that the paw reached in its first peak was statistically different in the innocuous versus painful stimuli (*Figure 4A*).

Next, we plotted the pain behavioral features that occur after the paw has reached its highest first peak and before the animal places its paw back to the surface. Similar to our observations with the four pre-paw peak pain behaviors, the four post-paw peak behaviors also show statistical separation between innocuous and pain stimuli (*Figure 4E–D*). Although we are not the first research group to observe paw shaking and guarding behaviors in rodents, this is one of the first technologies to automatically score the number and duration of these paw movement dynamics. Taken together, these automated measurements extracted from paw time series data are sufficient to objectively separate touch from pain in genetically diverse mice. Additionally, each one of our seven behavioral features was sufficient to independently separate touch versus pain during baseline conditions (*Figure 4*). Therefore, researchers can opt to choose any of these measurements to score the mouse pain state, and it's possible that some of these features will become more or less important depending upon the model used and the basal state of the mouse (*Figure 4—source data 1*).

## A univariate pain scale for scoring mice behavior

The seven behavioral features we extracted from video each provide more useful information than paw withdrawal frequency alone. Nevertheless, the simplest way to score behavior is using a univariate pain scale derived from these behavioral features. To accomplish this, we used ordinal logistic regression to identify a univariate linear subspace of the seven behavioral features that best separates the four somatosensory stimuli: CS, DB, LP, and HP. We did this for two cases: first, restricting to just four features quantifiable in the pre-peak period ($t < t^*$), and then for all seven features quantified during the post-peak period (*Figure 4H and I*, respectively). For both the pre- and post-peak paw features, the two no-pain stimuli (CS and DB) cover the same parts of the subspace and are largely indistinguishable. By contrast, the low and high pain conditions (LP and HP) clearly separate from the no-pain stimuli and also separate from each other in this univariate pain score.

The relative importance of each standardized (mean subtracted, variance scaled) behavioral feature was quantified as the loading of that feature on the univariate pain scale identified by ordinal logistic regression. The feature importance for both pre- and post-peak features (*Figure 4—figure*

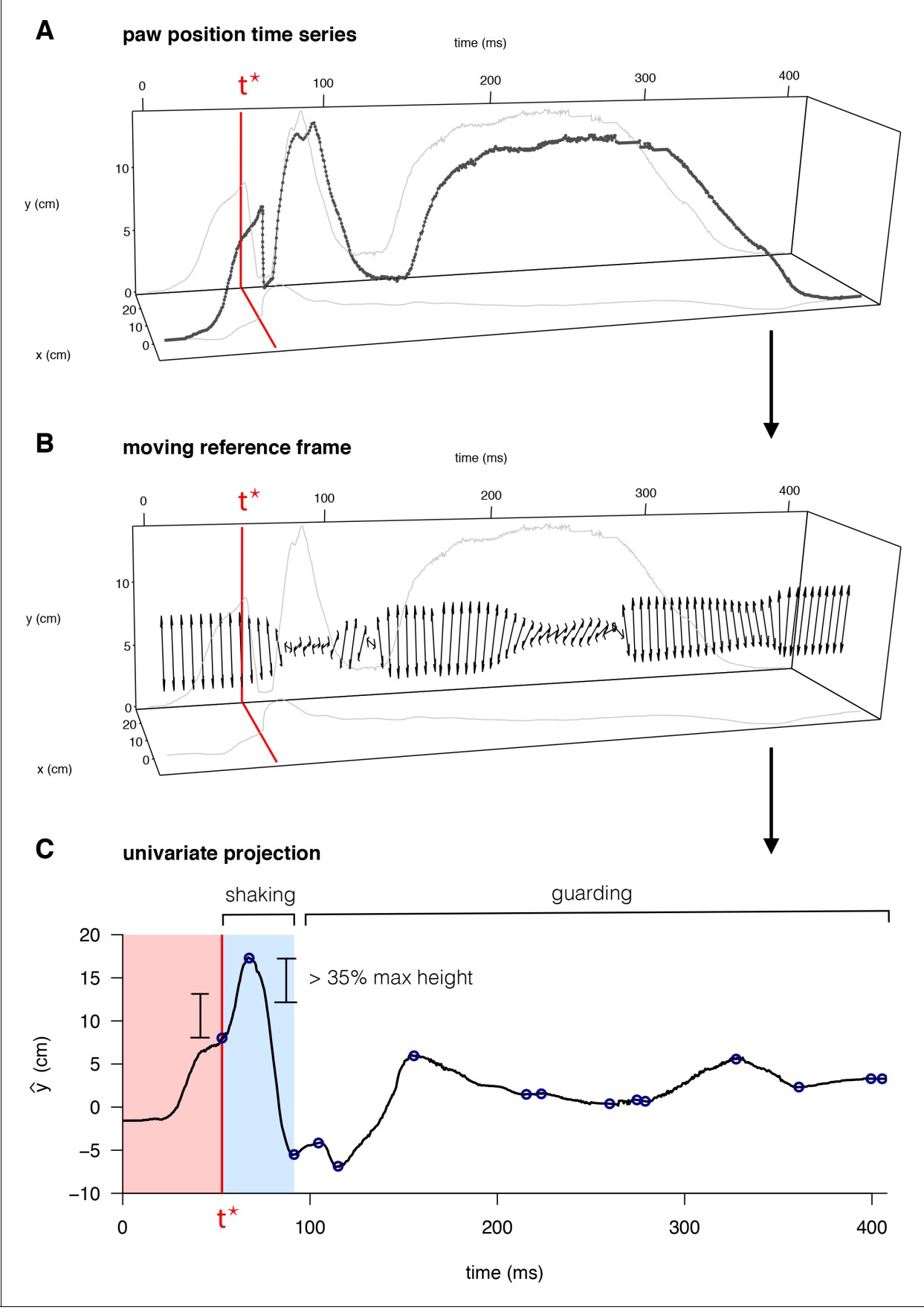

**Figure 3.** Quantification of behavioral features for mouse pain state. (**A**) Raw paw positions (x, y) measured in the camera reference frame (anterior/posterior displacement and vertical height, respectively) as a function of time, with the time of the first peak in paw height, t*, marked in red. (**B**) The principal axis of paw displacement in a moving time window (double-sided arrow), shown as a function of time. (**C**) Displacement in the principal axis moving reference frame. Local maxima and minima are indicated by blue circles – the first of which occurs at time t*. Periods of paw shaking were identified based on sequences of displacements between maxima and minima above a given threshold relative to the maximum paw height displacement. Guarding periods were defined simply as periods without shaking, in which the paw remained elevated above its final resting point.

supplement 1) indicate that relatively simple properties of the paw trajectory, such as the maximum height of the paw and the total distance traveled before the paw returns to resting position, contribute substantially to separating pain from no-pain. The max y velocity and the number of paw shakes further contributes to the separation of low versus high pain categories in the post-peak paw features (low and high pain classification is slightly improved by post- rather than pre-peak features, *Figure 4—figure supplement 2*).

Next, we used the PAWS system to quantify pain-related paw movement in mice treated with a common inflammatory pain agent, the complete Freund's adjuvant (CFA). To accomplish this, we used 20 mice (5 each of the C57BL/6, 129S1, Balb/c, and A/J strains chosen randomly) and applied the innocuous dynamic brush and the 4 g VFH filament before and 48 hr after unilateral hind paw injection of CFA. Since the dynamic brush is not painful and 4 g VFH lies close to the boundary separating painful and non-painful responses, we reasoned that we would observe a mechanical allodynia phenotype with increased PAWS values (*Abdus-Saboor et al., 2019*; *Alhadeff et al., 2018*). However, we did not observe a significant increase in the PAWS measurements following CFA (*Figure 4—figure supplement 3*). In fact, following CFA the responses to dynamic brush were slightly decreased, and the responses to 4 g VFH, although greater than to dynamic brush, nevertheless fell near the threshold separating touch and pain at baseline (*Figure 4—figure supplement 3*; *Figure 4—source data 2*). While performing these assays, we noted that the CFA-injected paw was red, inflamed, and swollen and the mice tried not to move the injured paw whatsoever. Thus, for peripheral manipulations that cause animals to drag heavy and swollen limbs, PAWS may not be a suitable system to detect pain hypersensitivity. Measuring thermal hyperalgesia with the Hargreaves assay, or detecting changes in mechanical threshold with VFHs, may be the most suitable method of quantifying CFA-induced pain hypersensitivity (*Petrus et al., 2007*; *Dhandapani et al., 2018*).

The performance of the univariate pain score was evaluated by leave-one-out (LOO) cross-validation for individual mice (*Figure 5A*), and by leave-one-strain-out for strains (*Figure 5B*). We compared the pre- and post-peak paw univariate pain scale classifications to a null model that assigns pain classes according to their probability in the training data, without reference to measured behavioral features. For prediction, CS and DB were treated as a single 'no-pain' class, while LP and HP were treated as a single 'pain' class, resulting in a binary classification: no-pain or pain (*Figure 5—figure supplement 1*). For predicting the pain state of a given mouse, or all the mice in a strain (generalizing across strains), post-peak paw features consistently performed best. In particular, the univariate pain score based on post-peak features was able to correctly identify the stimulus as painful or non-painful with 83.5% accuracy (cross-validated by leave-out one mouse); and it provided 81.3% accuracy when predicting the stimulus for a mouse of a novel strain (cross-validated by leave-out one strain). Nonparametric bootstrap 95% confidence intervals were [79.1%, 87.8%] and [76.9%, 85.9%], respectively.

## Applying automated tracking and PAWS to a new mouse line

After testing the seven inbred lines described above, we next tested an additional strain (SJL) that other labs have used in various experimental contexts. The SJL mice have not been studied in the context of pain, but some researchers use these mice to study aggression, as these mice display strong bouts of aggression and are prone to fight (*Lumley et al., 2004*; *Tellegen and Horn, 1972*). We reasoned that this strain would make a good test case for our new platform.

Using traditional scoring with the SJL strain, we noticed the response rate mirrored the canonical C57BL/6 strain (*Figure 1*), with low responsiveness to a cotton swab, and high responsiveness to the other stimuli (*Figure 6A*). Using this metric alone, we might have concluded that the SJL strain is no different than the most commonly used wildtype strain C57BL/6. However, when using high-speed videography and PAWS we find that SJL appears hypersensitive to mechanical pain – responding to

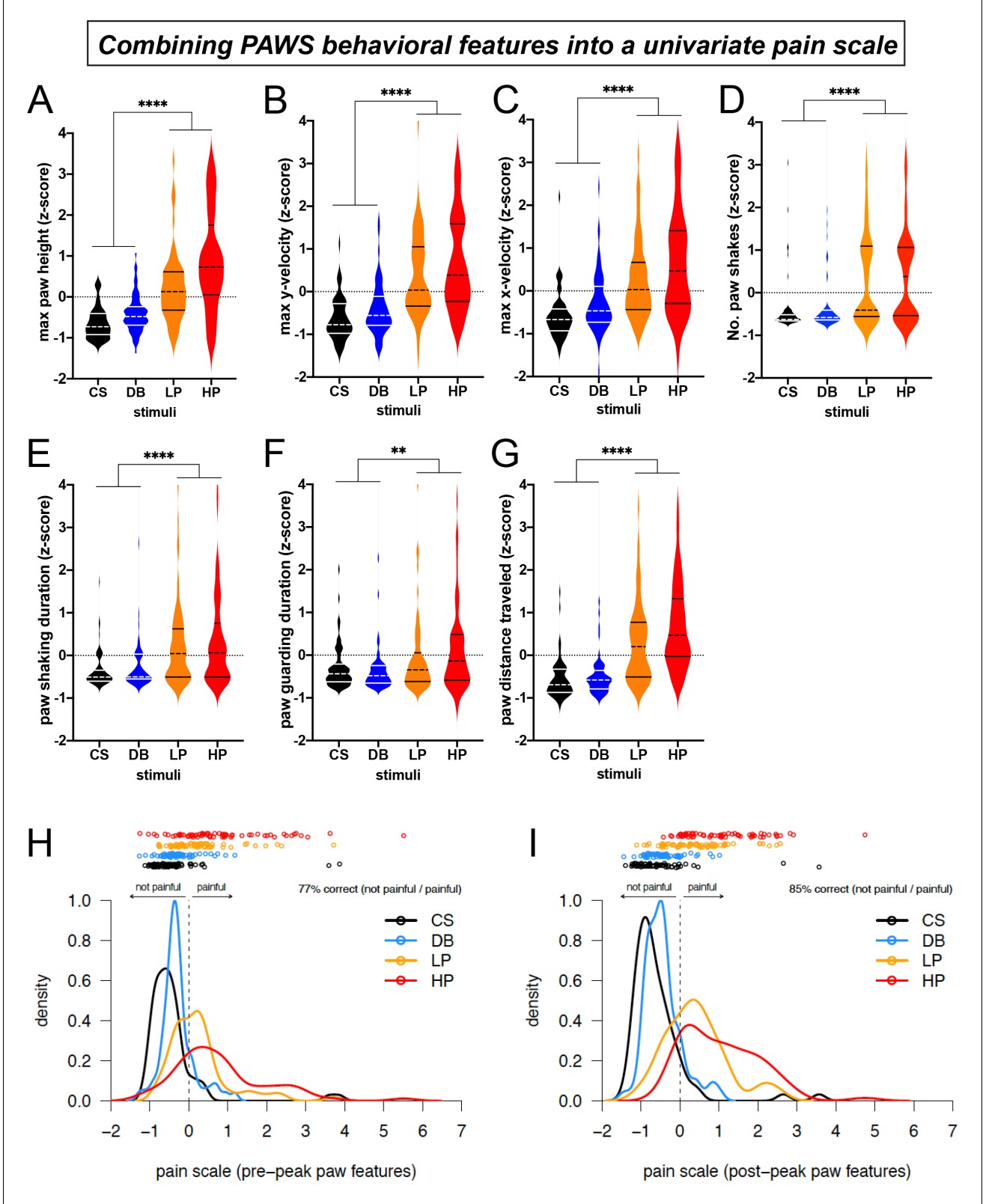

**Figure 4.** Automated measurement of pain behavioral features across the seven strains with new software PAWS. Measurements are converted to Z-score to reveal deviation from the mean of individual measures and to standardize units across the seven measures. (A-C) Pain measurements of the stimulated paw from lift to max height. (D-G) Pain measurements of the stimulated paw from max height to paw return. N = 10/mice of each strain given each stimulus once. Statistical significance was computed by comparing CS+DB versus LP+HP with Wilcoxon matched-pairs signed-rank test. **

*Figure 4 continued on next page*

*Figure 4 continued*

represents p-value ≤0.01. **** represents p-value ≤0.0001. On the violin plots, black or white horizontal lines represent quartiles and black or white dashed horizontal lines represent the median. (H) The univariate linear scale that best separates the four pain states studied: CS, DB, LP, and HP, for pre-peak features only. Kernel density estimates and the original data decomposed by pain state are shown projected onto the univariate measure of pain identified by ordinal logistic regression. Here, we use the threshold between no-pain and pain (DB and LP) inferred by ordinal logistic regression as the zero point, and we rescale by the threshold between low pain (LP) and high pain (HP), fixing this point at one. (I) Same as a but for post-peak paw features only.

The online version of this article includes the following source data and figure supplement(s) for figure 4:

**Source data 1.** Raw data values automatically computed from PAWS showing each mouse and stimulus across all eight inbred lines tested.

**Source data 2.** Raw data values automatically computed from PAWS showing each mouse and stimulus across four inbred lines tested before and after unilateral CFA paw injection.

**Figure supplement 1.** Relative feature importance for individual behavioral features that contribute to the univariate pain score.

**Figure supplement 2.** The univariate linear scale that best separates the four pain states studied: CS, DB, LP, and HP.

**Figure supplement 3.** Univariate linear scale using post-paw peak features before and after CFA injection and application of dynamic brush or 4 g von Frey hair filament.

the soft dynamic brush as if it were pinprick and having even more elaborate paw withdrawal stimuli (*Figure 6B*). The outlier pattern of this strain is observed when scoring the seven individual pain behavioral features. Consistent with this finding, our confidence in predicting the stimulus the animal received based on its response using our univariate pain scale is significantly reduced in the SJL strain (significant difference in mean LOO accuracy between DBA1 and SJL; one-sided Welch two-sample t-test, t = 2.9405, df = 36.894, p=0.003), reflecting the nature of the outlier phenotype in these mice (*Figure 6C*). These data reveal that our automated pain assessment platform is capable of detecting individual strain differences in susceptibility to mechanical pain and that this platform can provide new information that would not have been predicted based on simple paw withdrawal frequencies.

## PAWS detects motivational changes in pain perception following chemogenetic brain circuit manipulation

Finally, we sought to directly test whether our separation of pre- versus post-paw peak features (denoted by *t\**) could be functionally separated or attributed to nociception in supraspinal circuits for affective-motivational behaviors rather than circuits that might influence the sensitivity of the pre-peak reflexes. To accomplish this reasoned that we could not use a standard inflammatory or neu-ropathic pain model, as those assays would alter the pain circuit from periphery to the brain, and thus the question we sought to address in relation to separating pre- versus post-paw peak behaviors would be obscured. Therefore, we focused our attention on manipulating pain circuits in the BLA (*Corder et al., 2019*). Thus, we hypothesized that the post-paw peak features could be increased by selectively driving hyperactivity in a recently identified basolateral amygdala nociceptive ensemble. Briefly, we used the activity-dependent transgenic TRAP2 mice (*Fos*-FOS-p2A-iCre-ERT2) to gain viral-genetic access to BLA neurons that are responsive to a noxious pinprick to the left hind paw. The transgenic mice in combination with an AAV expressing an excitatory DREADD (AAV5-hSyn-FLEx-hM3q-mCherry) allow the specific expression of hM3 only in neurons responsive to the noxious pinprick stimulus (*pain*TRAP2$^{hM3}$; *Figure 7A*). We confirmed bilateral hM3-mCherry expression that was restricted to only the BLA and not within the neighboring central nucleus (*Figure 7—figure supplement 1*). We first performed our analysis on Cre-negative control animals and observed no behavioral effects of hM3-agonist CNO (i.p., 3.0 mg/kg). Next, we performed our behavioral analysis on *pain*TRAP2$^{hM3}$ mice at baseline (-CNO) and after activation of the BLA pain ensemble (+CNO) (*Figure 7*). Because activation of the BLA pain ensemble resulted in unilateral spontaneous guarding pain behaviors, we waited until mice were calm, stood still , and had all four paws on the surface before applying our sensory stimuli. We applied a cotton swab and dynamic brush on one day, and a light and heavy pinprick on a second day. We observed that the PAWS measurements at baseline (-CNO) mirrored those that we observed with the seven strains described above, showing separation between innocuous and noxious stimuli (*Figure 7B–H*). Conversely, with activation of the BLA nociceptive ensemble in *pain*TRAP2$^{hM3}$ mice (+CNO), we noticed increased separation with the post-paw peak pain measurements when delivering the heavy pinprick stimuli,

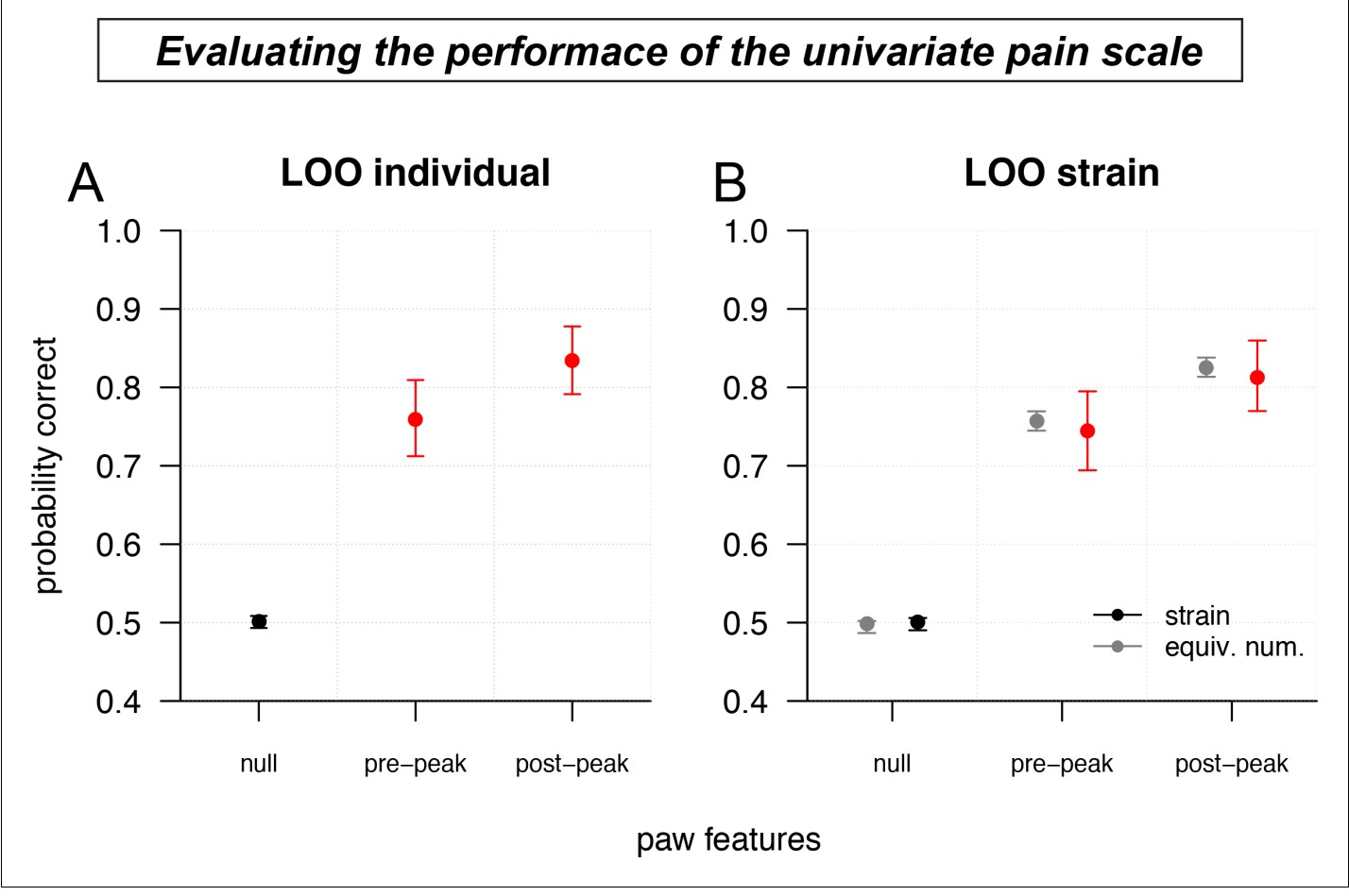

**Figure 5.** Validating the accuracy of the univariate pain scale. (**A**) Leave-one-out (LOO) cross-validation performance (binary classification accuracy of not pain/pain) of the pre-peak and post-peak univariate pain scales, compared to a null model (random assignment), for individual mice. Error bars show nonparametric bootstrap 95% confidence intervals. (**B**) Same as a, but leaving out an entire strain (colors) or an equivalent number of mice chosen uniformly at random (gray).

The online version of this article includes the following figure supplement(s) for figure 5:

**Figure supplement 1.** Validating the accuracy of the univariate pain scale.

most noticeable in the combined shaking/guarding measurement (*Figure 7H*). When we used ordinal logistic regression for statistical separation that combined our seven behavioral features, we observed an emergent behavioral representation of the post-paw peak features that was highly separated from controls, indicating that selective activation of supraspinal amygdalar ensembles shifted the internal state of the animals to engage in heightened nociceptive behaviors (*Figure 7I,J*). Specifically, after CNO administration all pain scale values with heavy pinprick applications are greater than every other stimulus (*Figure 7J*). Thus, PAWS automatically measures increased mechanical hyperalgesia to noxious stimuli that was influenced by top-down processes in the absence of peripheral or spinal sensitization. These data suggest that our pre- versus post-peak designations have biological relevance because we can genetically increase the gain on the more temporally-delayed behavioral representations, which may reflect the aversiveness of the pain experience (*Figure 7J*; *Figure 7— source data 1*).

## Discussion

Here, we describe an automated approach to quantify the most salient behavioral features following mechanical stimulation of the mouse paw for separating responses according to stimulus intensity.

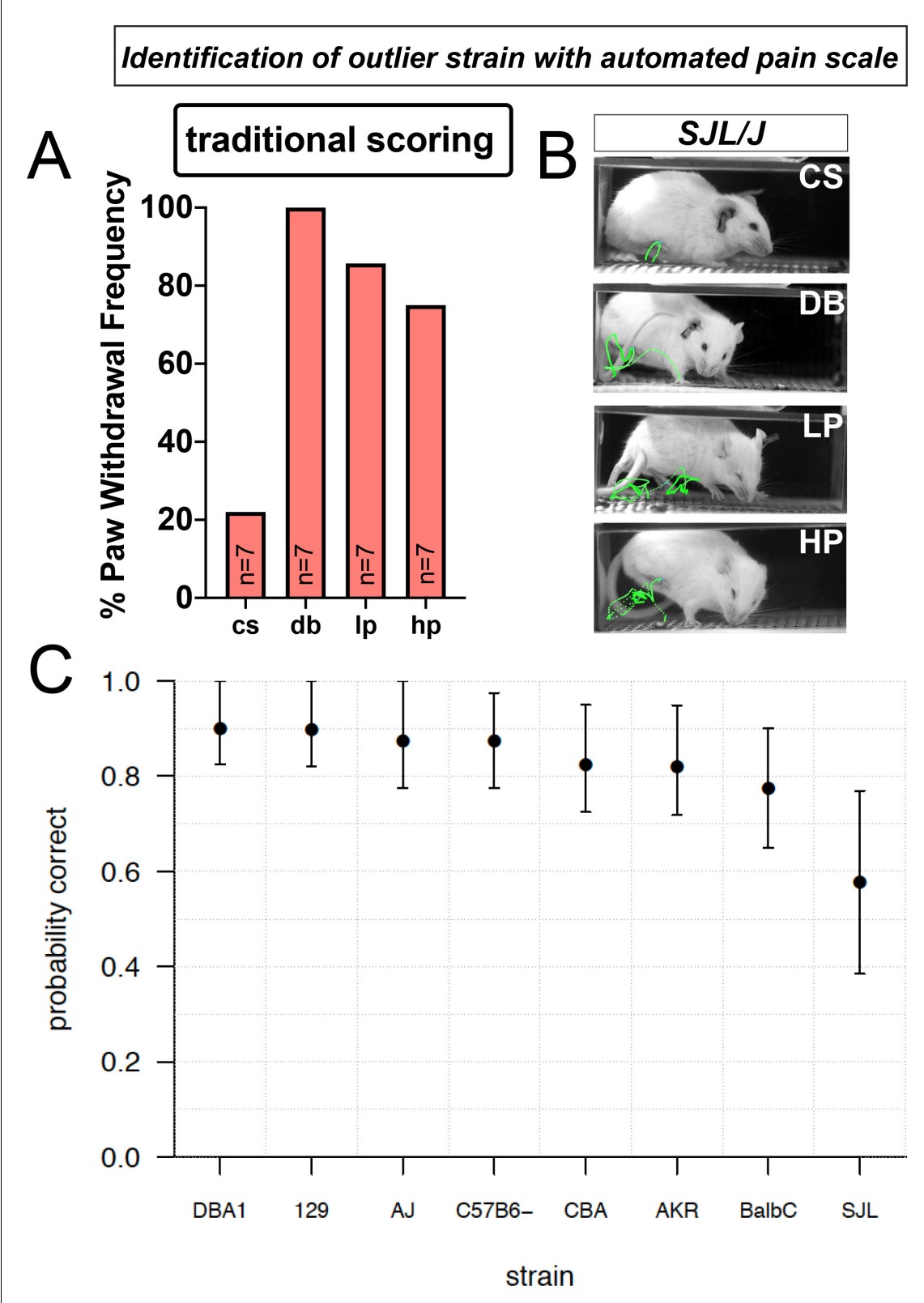

**Figure 6.** Automated pain assessment platform uncovers an outlier strain. (**A**) Traditional scoring focused on paw withdrawal frequencies to four mechanical stimuli: cs = cotton swab, db = dynamic brush, lp = light pinprick, hp = heavy pinprick. N = 7 mice for the SJL strain. (**B**) ProAnalyst tracking showing SJL mice have pain-like response to dynamic brush and heightened pinprick responses. (**C**) Leave-one-out (LOO) cross validation by strain

*Figure 6 continued on next page*

*Figure 6 continued*

shows stimulus prediction for SJL mice is poor, likely because their responses typically map outside of the normal range of the other seven strains. Error bars show nonparametric bootstrap 95% confidence intervals.

Scoring the paw withdrawal reflex to a natural stimulus is the most commonly used assessment method in preclinical rodent models with yes/no responsiveness used as a proxy for inferring pain states. While this methodology for measuring pain in rodents is not fundamentally flawed, it lacks resolution – a limitation that can now be overcome with advances in videography and automated tracking. Here, by automatically tracking paw dynamics at sub-second speeds with high-speed videography coupled with machine learning approaches, we reveal stereotyped trajectory patterns in response to innocuous versus noxious stimuli spanning genetically diverse mice. With an accurate pinpoint of the paw at high spatiotemporal resolution, a freely available software package we term PAWS automatically quantifies seven behavioral features that are combined to determine the mouse pain state. Notably, the seven behavioral features we defined have the benefit of quantifying pain response in terms of intuitive behaviors (such as paw shakes, guarding, etc), in contrast to generic unsupervised machine learning approaches. Demonstrating the robustness of the platform, we identify an outlier mouse strain that displays heightened pain sensitivity, and we accurately measure a heightened pain state when we simultaneously activate pain in the periphery and brain using chemogenetic and natural stimuli.

Behavioral neuroscience in model organisms is undergoing a renaissance with the emergence of new tools to automatically track and measure behavior (*Datta et al., 2019*; *Fried et al., 2020*; *von Ziegler et al., 2020*; *Günel et al., 2019*; *Graving et al., 2019*; *Itskovits et al., 2017*; *Kabra et al., 2013*; *Klibaite et al., 2017*). This renaissance is coincident with many in the research community questioning the robustness of rodent models of pain, addiction, depression, anxiety, and other neuropsychiatric disorders. Many researchers are taking steps backward to first properly understand the components of a complex behavioral sequence before proceeding to identify the neuronal correlates that drive those motor patterns (*Datta et al., 2019*; *Fried et al., 2020*; *Berman, 2018*). Both supervised and unsupervised machine learning algorithms are now able to follow unlabeled individual limbs on an experimental animal and automatically define behaviors of interest (*Wiltschko et al., 2015*; *Pereira et al., 2019*; *Mathis et al., 2018*; *Graving et al., 2019*; *Berman et al., 2014*; *Kabra et al., 2013*). Although the majority of these tools have yet to be adopted en masse by the pain research community, some of this technology is already in use by pain researchers. For example, the automated grimace scale developed by the Mogil and Zylka labs uses a convolutional neural network trained with 6000 facial images of mice in 'pain' or 'no-pain', to make accurate predictions of the mouse pain state in novel datasets (*Tuttle et al., 2018*). The automated grimace scale still requires additional customization to assess rodents of different coat colors and to measure chronic pain. Tools like the automated grimace scale that focus on the face, could be combined with the automated pain assessment platform described here that focuses on the paw, for a comprehensive picture of both evoked and spontaneous behavioral responsiveness.

Here, with our platform, we observed both homogeneties in behavioral responses across seven genetically distinct mouse lines, as well as an outlier strain with responses that mapped outside the range of those seven. A wealth of prior literature demonstrated that individual differences in responsiveness to pain in both mice and humans are driven in part by allelic variation in genes important for pain processing (*Mogil, 2012*; *Calvo et al., 2019*). In regards to the mouse, pioneering studies carried out 20 years ago by the Mogil group testing pain sensitivity across 11 inbred lines using 12 behavioral read-outs, revealed that depending upon the sensory modality tested, and whether the test was performed before or after injury to the somatosensory system, genotype appeared to influence mouse pain behaviors (*Mogil et al., 1999*). A meta-analysis from 10 years ago described over 400 papers from mouse pain research that implicated ~350 genes in pain and analgesia (*LaCroix-Fralish et al., 2007*). However, with the relatively limited resolution of some conventional pain behavior assays, and displayed here with our traditional scoring of the data, it remains unclear how reliable some of these studies are and which potential target genes merit further development as novel analgesics. This assertion is underscored by the fact that only a handful of targets that have shown promise in rodents, have made it to the clinic as novel therapeutics, causing many to question the robustness of the animal models used in pain testing (*Vardeh et al., 2016*; *Woolf, 2010*).

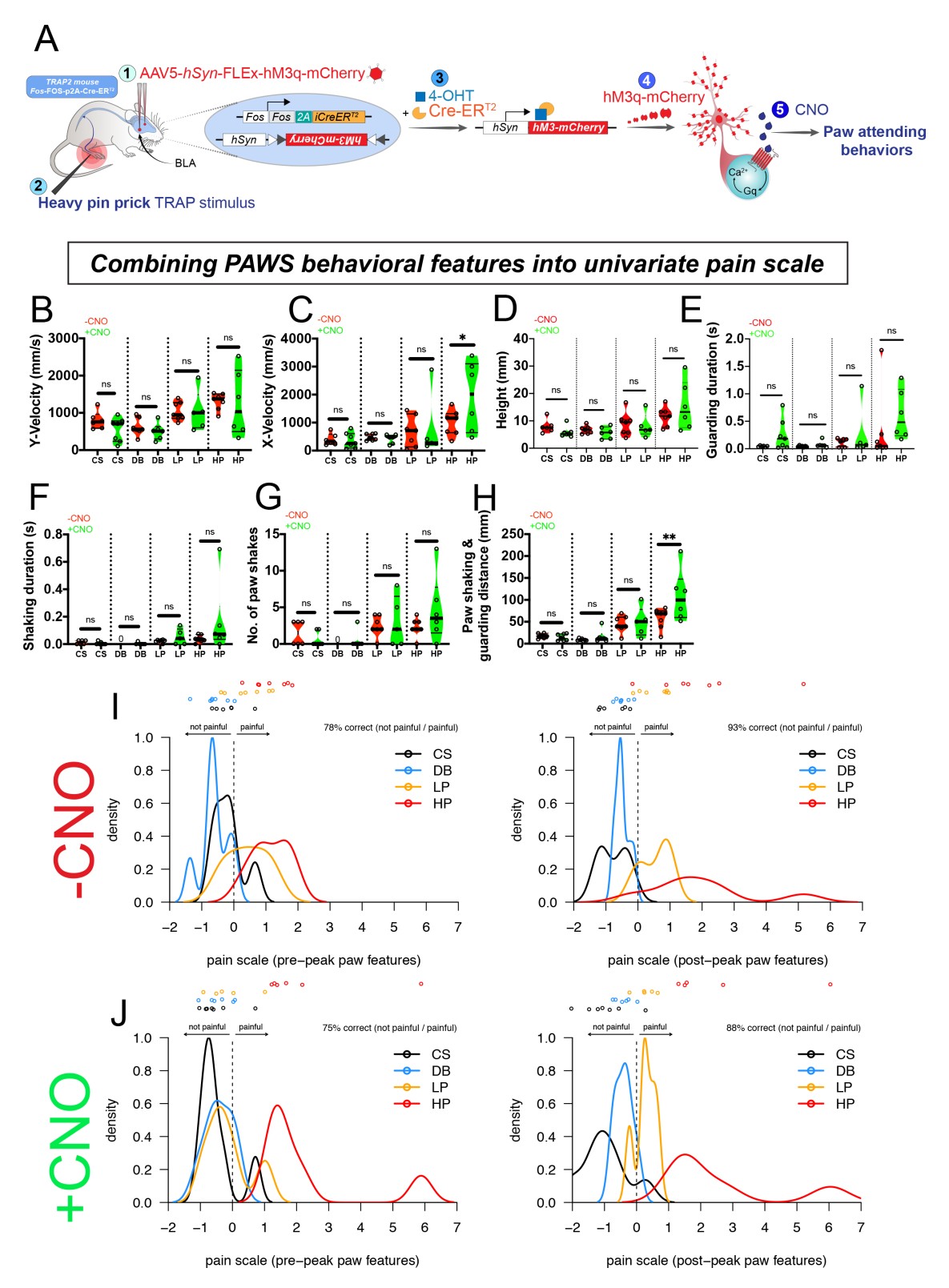

**Figure 7.** Pain hypersensitivity with chemogenetic activation of the BLA pain ensemble automatically captured via PAWS. (**A**) Schematic to permanently tag pain-active neurons in the BLA with an excitatory DREADD in transgenic *pain*TRAP2[hM3] mice. (**B-D**) Automatic measurement of pre-paw peak features comparing mice at baseline (-CNO) to mice administered CNO. (**E-H**) Automatic measurement of post-paw peak features comparing mice at baseline (-CNO) to mice administered CNO. Stimulus abbreviations are same as above. Statistical significance was computed with student's t-test. *p-

*Figure 7 continued on next page*

*Figure 7 continued*

value ≤0.05. **p-value ≤0.01. Raw values were plotted instead of z-scores as shown in *Figure 3* because we are plotting only six mice in this figure. (I-J) Ordinal logistic regression reveals separation of innocuous versus noxious stimuli at baseline (-CNO) and further segregation of the HP group after CNO, only when using all seven paw features. N = 6 mice tested in these experiments. The second panel in J omits one outlier mouse (HP treatment) that produced pain score >25.

The online version of this article includes the following source data and figure supplement(s) for figure 7:

**Source data 1.** Raw data values automatically computed from PAWS showing each *pain*TRAP2$^{hM3}$ mouse combined with sensory stimuli before and after CNO administration.

**Figure supplement 1.** Targeting excitatory DREADD in *pain*TRAP2$^{hM3}$ mice.

We also observed a new hypersensitivity phenotype, where SJL mice respond to a soft brush as if it were a pinprick. To the best of our knowledge, this is the first report of a pain hyper-sensitivity phenotype for SJL mice at baseline. Of note, SJL mice are known to be an aggressor mouse line and even in our studies, three animals had to be removed from testing due to excessive fighting between cage mates (*Miner et al., 1993*). Therefore, future studies are necessary to determine if the genetic repertoire that makes these mice aggressive also contributes to their heightened pain responses. Additionally, SJL mice are homozygous for the retinal degeneration one mutation, *Pde6b$^{rd1}$*, causing poor eyesight and even blindness in some of these mice (*Chang et al., 2002*; *Chang et al., 2013*). It is therefore possible that what we perceive as pain hypersensitivity may reflect more of heightened startle response, especially if the mice perceive an intense stimulation yet cannot localize who is delivering this stimulus. Future studies are warranted to determine if the altered behavioral changes seen here are driven by somatosensory system changes or are rather the byproduct of alterations elsewhere in the animal. Together, these results demonstrate that we can begin to use this platform as a behavioral screening tool to identify outlier strains (some will be of more interest than others), followed by subsequent genetic mapping approaches to uncover new alleles that may directly contribute to pain perception.

Finally, we tested the ability of our platform to uniquely identify changes that were driven by selective manipulation of a central pain circuit. We demonstrate that we can chemogenetically activate the BLA pain ensemble and detect hypersensitivity to peripheral stimuli. In addition to confirming the precision of our technology, these experiments raise an intriguing biological question: what is the emotional and sensory experience of activating pain-responsive neurons without a peripheral injury? Would tonic activation of this BLA pain ensemble be comparable to human experiences of chronic pain? Would DREADD inhibition of this BLA circuit be sufficient to block the post-paw peak behavioral features such as paw shaking and guarding? Further studies to examine these questions are ongoing. Additionally, we noticed that only the heavy pinprick stimulus increased the aversive responses when we reactivated the amygdalar nociceptive circuit. Based on our single-neuron micro-endoscope calcium imaging data in the basolateral amygdala (*Corder et al., 2019*), the nociceptive neurons respond only to strongly-noxious sensory stimuli. Only after peripheral nerve damage to induce chronic neuropathic pain, did we observe that formerly-innocuous stimuli engage this BLA ensemble. This suggests that some up-circuit plasticity occurred that redirected touch information into the BLA ensemble (e.g. opening of the spinal 'gate', see *Torsney and MacDermott, 2006*; *Cui et al., 2016*; *Petitjean et al., 2015*; *Braz et al., 2014*; *Cheng et al., 2017*; *Duan et al., 2014*; *Foster et al., 2015*). Thus, without this central plasticity, the other stimuli likely do not activate the ensemble under normal, uninjured conditions. Therefore, chemogenetic activation of the nociceptive ensemble is not necessarily predicted to amplify or modulate neural processes related to non-noxious stimuli. We did observe some spontaneous lifting and licking of the hind paws upon CNO treatment, which could reflect general aversive perception, but how this might alter withdrawal reflexes that use distinct neural circuits is unclear. The fact that we observed a robust increase in paw shaking and guarding-related behaviors supports the conclusion that the BLA nociceptive ensemble is specifically tuned to encode and modulate nociception only.

In summary, we have developed a rapid and user-friendly automated pain assessment platform for measuring mechanical pain in mice. Since mechanical stimulation of the rodent hind paw remains the most common method to measure pain in mice, the tools described here, with the addition of a high-speed camera, are fully compatible with the setups that most labs currently use. As such, we do not foresee major hurdles in the wide adoption of this methodology. To increase our fundamental

understanding of the neurobiology of pain and to translate our basic science findings in pain research to improved patient outcomes, the pain measurement tools in rodents must be robust. The platform described here should aid in that pursuit.

# Materials and methods

## Key resources table

| Reagent type (species) or resource | Designation | Source or reference | Identifiers | Additional information |
|---|---|---|---|---|
| Genetic reagent (*M. musculus*) | C57BL/6J | Jackson Laboratory | RRID:IMSR_JAX:000664 | |
| Genetic reagent (*M. musculus*) | A/J | Jackson Laboratory | RRID:IMSR_JAX:000646 | |
| Genetic reagent (*M. musculus*) | 129S1/SvlmJ | Jackson Laboratory | Stock No: 002448 | |
| Genetic reagent (*M. musculus*) | BALB/cJ | Jackson Laboratory | Stock No: 000651 | |
| Genetic reagent (*M. musculus*) | DBA/1J | Jackson Laboratory | Stock No: 000670 | |
| Genetic reagent (*M. musculus*) | AKR/J | Jackson Laboratory | RRID:IMSR_JAX:000648 | |
| Genetic reagent (*M. musculus*) | CBA/J | Jackson Laboratory | Stock No: 000656 | |
| Genetic reagent (*M. musculus*) | SJL/J | Jackson Laboratory | Stock No: 000686 | |
| Genetic reagent (*M. musculus*) | Fos-FOS-2A-iCre-ERT2 | Jackson Laboratory | Stock no: 030323 | PMID:28912243 |
| Chemical compound, drug | Clozapine N-oxide | Sigma | C0832 | 3 mg/kg |
| Chemical compound, drug | Complete Freund's Adjuvant | Sigma | F5881 | 10 µL/paw |
| Software, algorithm | PAWS | this paper | | https://github.com/crtwomey/paws |
| Software, algorithm | ProAnalyst | https://www.xcitex.com/proanalyst-motion-analysis-software.php | | |
| Software, algorithm | SLEAP | https://sleap.ai/ | | PMID:30573820 |

## Mouse strains

Mice for behavior testing were maintained in a barrier animal facility in the Carolyn Lynch or Translational Research Laboratory (TRL) buildings at the University of Pennsylvania. The Lynch and TRL vivariums are temperature controlled and maintained under a 12 hr light/dark cycle (7 am/7 pm) at 70 degrees Fahrenheit with ad lib access to food (Purina LabDiet 5001) and tap water. The feed compartment on the wire box lid of the cage was kept at a minimum of 1/3 full at all times. All cages were provided with nestlets to provide enrichment for mice. All procedures were conducted according to animal protocols approved by the university Institutional Animal Care and Use Committee (IACUC) and in accordance with the National Institutes of Health (NIH) guidelines. C57BL/6J, A/J, 129S1/SvlmJ, Balb/cJ, DBA/1J, AKR/J, CBA/J, SJL/J, and TRAP2 mice (Fos-FOS-2A-iCre-ERT2, stock no. 030323) mice were all purchased from Jackson Laboratories. All animals were habituated to our facility for 2 weeks after delivery before beginning behavioral experiments described below. All inbred strains of mice tested were male with 10 mice used per strain used unless otherwise noted in the figure or figure legend, while six female TRAP2 mice tested. The choice to use female mice in the TRAP2 experiments was random based on availability of mice. All mice were adults between 2 and 4 months. Animals were co-housed with 4–5 mice per cage in a large holding room containing approximately 500 cages of mice. For the SJL/J strain, some aggression was observed, and several mice had to be removed from testing and separated due to fighting.

## High-speed imaging and video storage

Mouse behaviors were recorded at 2000 fps with a high-speed camera (Photron FastCAM Mini AX 50 170 K-M-32GB - Monochrome 170K with 32 GB memory) and attached lens (Zeiss 2/100M ZF.2-mount). Mice performed behavior in rectangular plexiglass chambers on an elevated mesh platform. The camera was placed at a ~45° angle at ~1–2 feet away from the Plexiglas holding chambers on a tripod with geared head for Photron AX 50. CMVision IP65 infrared lights that mice cannot detect were used to adequately illuminate the paw for subsequent tracking in ProAnalyst. All data were collected on a Dell laptop computer with Photron FastCAM Analysis software. Each cohort of mice (n = 40) used roughly 80 GB of video storage space, (average size of video file = ~2 GB), which amounts to a total of ~160 GB for all eight inbred strains. The format for tracking data = . csv and. txt output (which amounted to ~5.4 MB for the eight inbred lines).

## Somatosensory behavior assays

In all behavioral experiments, we used a sample size of 6–10 mice per strain, as these numbers of consistent with studies of this kind in the literature to reach statistically significant conclusions. For experiments with inbred mice, animals were not ear-tagged, so we had no source of identification, and thus animals within a given strain were randomly tested. For experiments with fos-trap mice, both the experimenter and evaluator of the data were blind to which animals were in the -CNO versus +CNO experimental groups. In general, all mice were habituated for a minimum of 5 days, for one hour each day, in the Plexiglas holding chambers before testing commenced. During baseline, mice were tested in groups of five and chambers were placed in a row with barriers preventing mice from seeing each other. On testing day, mice were habituated for an additional ~10 min before stimulation and tested one at a time. Stimuli were applied through the mesh to the hind paw proximal to the camera. Testing only occurred when the camera's view of the paw was unobstructed. Mice only received one stimulus on a given testing day (cs, db, lp, or hp) and were given at least 24 hr between each stimulus session. Stimuli were tested from least painful to most: cotton swab, dynamic brush, light pinprick, and heavy pinprick. Cotton swab tests consisted of contact between the cotton Q-tip and the hind paw until paw withdrawal. Dynamic brush tests were performed by wiping a concealer makeup brush (L'Oréal Paris Infallible Concealer Brush, item model number 3760228170158) across the hind paw from back to front. Light pinprick tests were performed by touching a pin (Austerlitz Insect Pins) to the hind paw of the mouse. The pin was withdrawn as soon as contact was observed. Heavy pinprick tests were performed by sharply pressing this pin into the paw so that it was pushed upward, without the breaking the skin barrier. The pin was withdrawn as soon as approximately 1/3 of the pin's length had passed through the mesh. For the application of von Frey hairs (VFHs, Stoelting Company, 58011), we used 4 g of force. As previously described, the VFH was directed at the center of the plantar paw and pressed upward until the filament bent (*Abdus-Saboor et al., 2019*). For inducing inflammatory pain, approximately 10 µL of Complete Freund's Adjuvant (CFA, Sigma, F5881) was injected into the plantar surface of 3% isoflurane-anesthetized mice as previously published (*Abdus-Saboor et al., 2019*).

## Automated paw tracking

We used ProAnalyst software to automatically track hind paw movements following stimulus application. This software allowed us to integrate automated and manually scored data, possible through the 'interpolation' feature within ProAnalyst. We were able to define specific regions of interest (paw), track, and generate data containing 'x' and 'y' coordinates of the paw through time, as well as velocity, speed, and acceleration information. In a subset of videos, additional manual annotation was performed for increased accuracy. For deep learning-based paw tracking with the SLEAP algorithm, we pseudo-randomly chose a small set of training frames from each video and hand-labeled the paw and toe. We trained SLEAP to predict toe and paw positions in unlabeled video frames (>~90% of total video frames). To generate trajectories, we overlaid the inferred x, y positions of the toe and paw in each video frame on a single still image corresponding to the apex of the mouse's paw during the assay.

## Development of PAWS software to quantify pain behaviors

Behavioral features were extracted from raw paw position time series in an automated and standardized procedure. First, the start and end of paw movement (paw at rest on the ground) were identified, and analysis was restricted to this time window. Peaks in paw height were then determined based on Savitsky-Golay smoothed estimates of paw velocity, and the first peak identified. The time of the first peak (designated t*) was used to separate pre-peak behavioral feature calculations from post-peak calculations. To differentiate shaking from guarding in the post-peak period, we constructed a moving reference frame based on the principal axis of paw displacement across a sliding window (0.04 s in duration) for each time point, and identified periods of consecutive displacements above a specified threshold (35% of maximum paw height) as periods of shaking. Note that in the construction of the moving reference frame the principal axes of variation were recovered via principal component analyses, which is not invariant to the sign of the recovered axes. Since displacement is measured over time it is sensitive to reversals in sign along the axis we measure it. We therefore ensured consistency by using the axis direction minimizing the angular deviation from the axis recovered at the previous time step. PAWS is open source and freely available at https://github.com/crtwomey/paws.

## Drugs

4-hydroxytamoxifen (Hello Bio, #HB2508) prepared in Kolliphor EL (Sigma, #27963), Clozapine-N-oxide (Hello Bio, #HB6149), and 0.9% sodium chloride (Sigma, #S3014).

## Viral reagents

For chemogenetic manipulation of BLA pain-active neurons, we intracranially injected 200 nL of AAV5-$hSyn$-$DIO$-$h$M3D($G_q$)-$mCherry$ (Addgene, titer: $7 \times 10^{12}$) into both the left and right BLA at coordinates AP: $-1.4$ mm, ML:$\pm3.1$ mm, DV: $-4.2$.

## Stereotactic injections and surgical procedures

We conducted surgeries under aseptic conditions using a small stereotaxic instrument (World Precision Instruments). We anesthetized mice with isofluorane (5% induction, 1–2% maintenance) during the entire surgery and maintained body temperature using a heating block. We injected mice with a beveled 33G needle attached to a 10 µL syringe (Nanofil, WPI) for delivery of 200 nL of viral reagent at a rate of 40 nL/min. After viral injection, the needle remained at the injection depth for 10 min before slow withdrawal over 2 min. After surgery, we maintained the animal's body temperature using a heating pad.

## Targeted recombination in active populations (TRAP) of BLA pain ensemble

We utilized female Fos[2A-iCreERT2] (TRAP2) mice, Jackson Laboratory, stock #030323 aged P46-73. We bilaterally transduced an AAV (AAV9-hSyn-DIO-hM3D(Gq)-mCherry) into the BLA. Two weeks after injection, mice were stimulated with a noxious pinprick on the left hind paw every 30 s for 10 min. One hour later, we injected mice with 4-hydroxytamoxifen (4-OHT) (20 mg/kg in 0.2 mL vehicle; subcutaneous) to induce genetic recombination. Eight weeks following 4-OHT administration, we examined mouse behavior. We examined behavior 30 min after injection of CNO (3 mg/kg in 0.2 mL vehicle; subcutaneous) or vehicle. Mice were euthanized via transcardial perfusion 4 months after viral injections. Formalin-fixed brains were collected and sectioned at 50 µm on a cryostat. Tissue was mounted and imaged on a fluorescent Keyence microscope.

## Acknowledgements

We thank Dr. Mala Murthy and Nat Tabris for the development of the deep neural network SLEAP and freely sharing an early and unpublished version of this platform with us. We thank members of the Abdus-Saboor, Plotkin, Murthy, and Corder labs for helpful comments on this manuscript. IA-S, JJ, and WF are supported by startup funds from the University of Pennsylvania and by a grant from the National Institutes of Health (NIH/NIDCR, R00-DE026807). OMA and TDP are supported by NIH BRAIN Initiative R01 NS104899. OMA is also supported by the BWF PDEP. Members of the Corder

lab are supported by NIH/NIDA grant R00-DA043609. JBP is supported by the Defense Advanced Research Projects Agency NGS2 program (grant D17AC00005), the Army Research Office (grant W911NF-17-1-0083), and the David and Lucile Packard Foundation. CRT is supported by the UPenn mindCORE fellowship.

## Additional information

### Funding

| Funder | Grant reference number | Author |
|---|---|---|
| National Institutes of Health | R00-DE026807 | Jessica M Jones Ishmail Abdus-Saboor |
| National Institutes of Health | R01 NS104899 | Osama Ahmed Talmo D Pereira |
| National Institutes of Health | R00-DA043609 | Jessica A Wojick Gregory Corder |
| Army Research Office | W911NF-17-1-0083 | Joshua B Plotkin |
| Defense Advanced Research Projects Agency | D17AC00005 | Joshua B Plotkin |
| Burroughs Wellcome Fund | PDEP | Osama M Ahmed |
| mindCORE | UPenn mindCORE fellowship | Colin Twomey |
| David and Lucile Packard Foundation | | Joshua B Plotkin |

The funders had no role in study design, data collection and interpretation, or the decision to submit the work for publication.

### Author contributions

Jessica M Jones, William Foster, Conceptualization, Resources, Data curation, Software, Formal analysis, Validation, Investigation, Visualization, Methodology, Writing - review and editing; Colin R Twomey, Conceptualization, Resources, Software, Methodology, Writing - original draft, Writing - review and editing; Justin Burdge, Conceptualization, Resources, Data curation, Formal analysis, Investigation, Visualization, Methodology, Writing - review and editing; Osama M Ahmed, Resources, Investigation, Methodology, Writing - review and editing; Talmo D Pereira, Investigation, Methodology, Writing - review and editing; Jessica A Wojick, Data curation, Formal analysis, Investigation, Writing - review and editing; Gregory Corder, Data curation, Formal analysis, Investigation, Methodology, Writing - review and editing; Joshua B Plotkin, Conceptualization, Resources, Data curation, Software, Formal analysis, Supervision, Validation, Investigation, Visualization, Methodology, Writing - original draft, Project administration, Writing - review and editing; Ishmail Abdus-Saboor, Conceptualization, Resources, Data curation, Formal analysis, Supervision, Funding acquisition, Validation, Investigation, Visualization, Methodology, Writing - original draft, Project administration, Writing - review and editing

### Author ORCIDs

Jessica M Jones  https://orcid.org/0000-0003-3638-255X
Talmo D Pereira  http://orcid.org/0000-0001-9075-8365
Joshua B Plotkin  http://orcid.org/0000-0003-2349-6304
Ishmail Abdus-Saboor  https://orcid.org/0000-0003-2120-0063

### Ethics

Animal experimentation: This study was performed in strict accordance with the recommendations in the Guide for the Care and Use of Laboratory Animals of the National Institutes of Health. All of the animals were handled according to approved institutional animal care and use committee (IACUC) protocols (#806519) of the University of Pennsylvania.

#### Decision letter and Author response

Decision letter https://doi.org/10.7554/eLife.57258.sa1
Author response https://doi.org/10.7554/eLife.57258.sa2

## Additional files

### Supplementary files

• Transparent reporting form

### Data availability

Raw data are associated with figures as source data.

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
