## [Decision Letter]

**Decision letter after peer review:**

[Editors’ note: the authors submitted for reconsideration following the decision after peer review. What follows is the decision letter after the first round of review.]

Thank you for submitting your work entitled "A machine-vision approach for automated pain measurement at millisecond timescales" for consideration by *eLife*. Your article has been reviewed by three peer reviewers, and the evaluation has been overseen by a Reviewing Editor and a Senior Editor, one of whom is a member of our Board of Reviewing Editors. The following individuals involved in review of your submission have agreed to reveal their identity: Rebecca Seal (Reviewer #1); Andrew Shepherd (Reviewer #2).

Our decision has been reached after consultation between the reviewers. Based on these discussions and the individual reviews below, we regret to inform you that your work will not be considered further for publication in *eLife*.

Specifically, while an automated, objective approach to measure evoked mechanical pain behavior in rodents will be a highly significant contribution to the pain field and the authors have presented solid efforts toward this goal, several critical aspects of the PAWS system and its validation that would successfully elevate the approach to a level above the manual one published previously (Abdus-Saboor et al., t) remain underdeveloped or missing. (1) A clear, well-defined and easy to understand method for reporting the response using PAWS is lacking. (2) The point of measuring SJL and 129S1 strains as extremes using PAWS was weakened by the absence of measuring traditional von Frey and also motor function. (3) While the reviewers understand the appeal of the eDREADD approach to identify the role of specific brain circuits in pain, a classic inflammatory pain model to demonstrate the detection of hypersensitivity by PAWS would be more straightforward and have broader utility. (4) Many of the quantitative aspects of foundational measures used to interpret PAWS were lacking, for example, how non-overlapping response is defined in the LDA and statistical analysis for interpretations of significance. Please also see the full Comments from the Reviewers, included below. We do appreciate that you could make revisions that would address some of the reviewers' specific comments, but our determination is that addressing these concerns would be beyond the scope of a revision at *eLife*.

Reviewer #1:

This work described in the manuscript by Jones, Foster, et al., describes an automated protocol to quantify and interpret (based on 8 components of the paw trajectory: 4 pre and 4 post) the response of mice to noxious and innocuous mechanical stimuli applied to the plantar hind paw using high speed video recordings. The protocol is designed to objectively report whether the mouse interprets the stimuli as non-painful or painful as well as the severity or intensity of the pain. Mouse strains that show extremes in their response to the stimuli (hypo and hyper) and also eDREADD activation of the pain neuronal ensemble within the amygdala were used to validate whether the protocol is able to accurately quantify and interpret a range of stimuli.

Comments:

The development of an easily accessible, automated and objective way to measure evoked mechanical pain in rodents will be extremely valuable for the field, as will delineating affective versus discriminative components (and in the context of chronic pain) and this is a good first step.

1) Authors should discuss variability in PAWs in relation to the variability inherent in the methods used to simulate the paw.

2) PAWS data in Figure 4 are showing significant differences between CS+DB and LP+HP in the scoring but the LD1/LD2 analyses extract differences in the LP and HP for post-peak features. A deeper explanation of how data are to be reported be interpreted would be helpful.

3) Activation of the pain ensemble with the DREADD and CNO evokes a pain like behavior. I would have expected that the LP would have resulted in greater shaking and guarding behavior than the -CNO control. Do the authors have an explanation?

4) Authors should discuss the interpretation for how mice like the CBA strain, which seem hypersensitive in the traditional testing method but are apparently within range by PAWS, compare to the SJL mice, which were not tested by the traditional method but are deemed hypersensitive by PAWS.

5) The 0.7 probability correct shown in Figure 5 and Figure 6 as being sufficient should be interpreted for the reader a bit more than what is provided in subsection “Automated scoring of rapid paw dynamics and lingering pain behaviors”. Related to this: There are no stats for these comparisons.

6) The rationale for mentioning a potential relationship to aggressive behavior in the Discussion section is not well-developed. What about a relationship of PAWS outcomes to the magnitude of a startle reflex?

Reviewer #2:

In this article, Jones et al. describe an automated assessment of tactile sensitivity. They report development and validation of a novel combination of high-speed videography and automated paw tracking. With this resource, they demonstrate that paw withdrawal to innocuous versus noxious stimuli can be separated in six inbred mouse strains. Using this system, they also present evidence that activating an ensemble of basolateral amygdala neurons during noxious stimulation changes paw withdrawal metrics. This approach represents a significant advance in behavioral assessment of tactile sensitivity, with the potential to contribute to much-needed discoveries in this domain.

Major comments:

This manuscript builds upon prior machine vision-based approaches in other animal models. High-speed videography of such withdrawal responses has been attempted before, but the demonstration of the ability to detect chemogenetic activation of pain aversion neurons in the amygdala and the validation across multiple inbred strains are new developments. The importance of this approach stems from its potential to address previously challenging problems, i.e. high-throughput, objective assessment of sensory and affective components of innocuous touch and noxious stimuli. PAWS has the potential to drastically improve the resolution and dimensionality of rodent pain scoring when compared to subjective 'yes/no' withdrawal scoring and threshold calculation, which has been the standard in the field for several decades.

1) Chemogenetic activation of a BLA pain ensemble is a worthwhile experiment, but the rationale for choosing this particular experiment wasn't particularly clear in the manuscript. Many potential experiments could have been proposed that would target some aspect of the pain neuraxis, with the hypothesis that a shift in sensitivity would be detected by the PAWS system. Why did the authors opt for chemogenetic activation of a BLA ensemble? Furthermore, would an equivalent experiment using an inhibitory DREADD have been likely to tell us anything further? Can these experiments tell us anything about the affective component of a response to a noxious stimulus in terms of paw withdrawal responses?

2) It would benefit the readers to state more clearly that analgesic reversal of pain-related changes to paw withdrawal were detected in Abdus-Saboor et al., (2019), wherein a software-assisted, manual scoring system of similar indices was used. Reversal of changes in paw withdrawal-associated behaviors is a robust indicator that they are pain-related.

3) In Figure 6, the authors mention two outlier strains (129S1 and SJL), which exhibit unusually low and high degrees of sensitivity. I applaud the authors for the inclusion of these datasets; it benefits the research community to describe the limitations and caveats of a new assay as early and as comprehensively as possible. However, are these shifts in sensitivity in 129S1 and SJL seen with conventional von Frey hair testing, as reported for the other strains in Figure 1B?

Whether or not von Frey data 'match' the PAWS data for 129S1 and SJL mice, it is likely to be instructive. A match between von Frey and PAWS data would be further validation of the accuracy of the PAWS system, and a discrepancy would raise the possibility that PAWS is extracting information from a multidimensional dataset which cannot be achieved with conventional von Frey assessment.

4) Related to the previous point, it is not clear if prior reports characterizing pain sensitivity in 129S1 versus other strains have also seen relative hyposensitivity. The discussion does mention that such strain differences have been assessed, but no direct comparisons are between these reports and the data in this manuscript are made.

5) In subsection “Statistical modeling with linear discriminant analyses separates touch versus pain across six inbred mouse strains” the authors note that pre-peak paw movements are nocifensive in nature, whereas those behaviors seen post-peak (shaking/guarding) are supraspinal in origin. I agree that this distinction seems plausible, given the timescales involved, but are there data in the literature to support this? Unless this is founded upon findings from prior studies, this comment might be more at home in the Discussion section.

Reviewer #3:

While I appreciate the importance of what the authors are seeking to accomplish in this study, it is not obvious how PAWS (Pain Assessment at Withdrawal Speeds), which scores eight defined behavioral endpoints, can be easily used by researchers in the pain field to quantitatively evaluate the magnitude of pain or to accurately predict the pain state. The authors convincingly show that a painful pinprick evokes a withdrawal response that looks very different from a non-painful response in terms of trajectory and pattern. A more straightforward approach would thus be to develop an algorithm that captures and classifies these trajectory and pattern differences over time. This would likely generate a more accurate and simple classifier relative to what the authors did-breaking these distinct motions into component variables like X and Y velocity, paw height, etc. The authors should seek to capture this visual difference in paw withdrawal trajectories with a single metric, and then show that this metric is scalable based on pain intensity. Without such a measure, the current implementation of PAWS is of limited general use for those who are interested in studying pain in mice. For example, what endpoint(s) is someone in the field supposed to use when studying pain and responses to analgesics? Y-velocity? X-velocity? # paw shakes? Paw height? All of these variables? Two of these variables? Three? Four? It is confusing and not simplified.

The title of the paper implies the machine vision approach is automated; however, this does not appear to be the case. The first step in this process is to manually label the center of the stimulated paw. This is thus more akin to a "semi-automated" approach.

Time of the first withdrawal peak (t*) is a critical variable in their analytic pipeline. However, it is unclear precisely how t* is calculated or defined. In the paper, authors write that t* is the time leading up to the initial paw peak. And in Figure 3A, since this is a 3d graph in 2d, it is unclear where t* is relative to the trajectory data.

From the LDA analyses, the authors state that the low and high pain stimuli separate from the no-pain stimuli (Figure 5A,B). However, I do not see a clear separation between these groups in the figure. Instead there appears to be significant overlap, which raises the question as to how specific the LDA analysis is at discriminating, in a quantitative manner, the magnitude of a pain response.

Figure 1B. The paw withdrawal frequency data has no error bars and the number of mice used to generate these data is not indicated. The authors write that strains differ in some of these assays but provide no statistics to confirm that the differences shown are statistically significant. Moreover, the authors are encouraged to consult and cite work by Jeff Mogil's group who evaluated mechanical sensitivity in different mouse strains many years ago.

Figure 3B and C are difficult to interpret. Provide more details in legend and in the figure itself. Ex. in B, are those lines with two arrow heads? What does the length of the line and angle of the line mean? And in C, how was the shaking vs guarding bout determined? Was this done by a human or did the algorithm make these assessments in an unbiased manner?

Figure 6. Authors state 129 and SJL mice are outliers, but based on data presented in this figure, it is hard to appreciate how exactly they are outliers. In panel B, 129 mice show a similar probability correct relative to all other strains, and SJL error bounds largely overlap the other strains. An outlier is typically defined as being two or more standard deviations from the mean.

Moreover, the atypical withdrawal response may have nothing to do with a pain hyposensitivity phenotype, as the authors assert. Instead, these strains may simply have motor deficits that prevent them from performing more vigorous/elaborate paw withdraw responses.

The use of chemogenetic amygdala stimulation to demonstrate the efficacy of PAWS for detecting hypersensitivity to noxious stimuli seems out of place in this study. A simpler, more straightforward, and more broadly applicable (for pain field) approach would be to inflame hindpaw with complete freunds adjuvant and/or perform a nerve injury surgery. These are commonly used ways of inducing pain hypersensitivity in the field, and hence as a first test, it will be important to show that PAWS can detect this form of hypersensitivity.

Moreover, it will be important to show that PAWS can detect graded changes in pain hypersensitivity, such as in response to a known analgesic.

---

## [Author Response]

[Editors’ note: The authors appealed the original decision. What follows is the authors’ response to the first round of review.]

Specifically, while an automated, objective approach to measure evoked mechanical pain behavior in rodents will be a highly significant contribution to the pain field and the authors have presented solid efforts toward this goal, several critical aspects of the PAWS system and its validation that would successfully elevate the approach to a level above the manual one published previously (Abdus-Saboor et al., t) remain underdeveloped or missing.(1) A clear, well-defined and easy to understand method for reporting the response using PAWS is lacking.

We want to thank the referees for this suggestion – which has led us to completely rework the statistical basis of the PAWS analysis and software, focused on producing a simple univariate pain scale.

To achieve this, we have completely removed the two-dimensional linear discriminant analysis (LD1/LD2). Instead, we develop a simpler, univariate measure of pain intensity that can be automatically scored from the motions tracked in the videography. The technical term for the underlying statistical method is an ordinal logistic regression. The regression is still based on underlying features of paw movement (X and Y velocity, max paw height etc) extracted automatically from the videography – but the upshot is a simple univariate measure of pain intensity.

Using a single, univariate measure has tremendously simplified our analyses and made the results easier to interpret. And we find that this simple univariate pain scale is even slightly better at discriminating between pain and no-pain stimuli as compared to the more complicated, two-dimensional analysis in our original submission.

Once again, we thank the referee for pushing us in this direction, which has led to a different implementation of PAWS that is easier to interpret and to generalize.

(2) The point of measuring SJL and 129S1 strains as extremes using PAWS was weakened by the absence of measuring traditional von Frey and also motor function.

This is a wonderful point and we have now added the traditional scoring method with the SJL strain (Figure 6A) and the 129S1 strain (Figure 1B). This clearly shows that using the PAWS system provides more information about pain sensitivity than the traditional method. Moreover, another referee pointed out, strain 129 was not statistically different than other strains in our original analyses. We have revisited this question in the revised analyses, using a univariate pain scale. And the referee is correct: 129 is not statistically different from other strains in our experiments and analyses, as reflected in the updated manuscript. (Whereas strain SJL remains a strong outlier in pain hypersensitivity).

(3) While the reviewers understand the appeal of the eDREADD approach to identify the role of specific brain circuits in pain, a classic inflammatory pain model to demonstrate the detection of hypersensitivity by PAWS would be more straightforward and have broader utility.

We appreciate this point and in the text we have clarified why we needed to use an excitatory DREADD approach with fos-trap2 mice to address our question. We were not simply trying to determine if PAWS can detect increasing pain levels, as we have already demonstrated this earlier in the paper. We were attempting to determine if PAWS could detect behaviors driven specifically by central input and causing a peripheral injury with a classic inflammatory model would obscure this goal.

Nonetheless, the reviewers comments about the utility of seeing how PAWS performs with a common inflammatory pain model are valid, and thus we performed these assays with 10 C57BL/6 mice. Admittedly, as described below and now included in the main text of the manuscript with accompanying supplemental figure, we did not observe the expected results. In an attempt to be fully transparent and not hide a negative result, we would like to publish these findings as a supplemental figure and we hope the reviewers agree that this is appropriate. We presume that a spontaneous pain detection using deep learning approaches could be valuable with the CFA model, but such a project is beyond the scope of this work. Nonetheless, the PAWS platform presented here, in our opinion, still represents a major leap forward in measuring peripherally and centrally driven mechanical pain in mice without inflamed paws.

New text:

“Next, we used the PAWS system to quantify pain-related paw movement in mice treated with a common inflammatory pain agent, the complete Freund’s adjuvant (CFA). To accomplish this, we used 20 mice (5 each of the C57BL/6, 129S1, Balb/c, and A/J strains chosen randomly) and applied the innocuous dynamic brush and the 4g von Frey Hair (VFH) filament before and 48 hours after unilateral hind paw injection of CFA. Since dynamic brush is not painful and 4g VFH lies close to the boundary separating painful and non-painful responses, we reasoned that we would observe a mechanical allodynia phenotype with increased PAWS values [10, 19]. However, we did not observe a significant increase in the PAWS measurements following CFA (Figure 4 —figure supplement 3). In fact, following CFA the responses to dynamic brush were slightly decreased, and responses to 4g VFH, although greater than to dynamic brush, nevertheless fell near the threshold separating touch and pain at baseline (Figure 4 —figure supplement 3) (Figure 4—source data 2). While performing these assays we noted that the CFA-injected paw was red, inflamed, and swollen and the mice tried not to move the injured paw whatsoever. Thus, for peripheral manipulations that cause animals to drag heavy and swollen limbs, PAWS may not be a suitable system to detect pain hypersensitivity. Measuring thermal hyperalgesia with the Hargreaves assay, or detecting changes in mechanical threshold with VFHs, may be the most suitable method of quantifying CFA-induced pain hypersensitivity [20, 21].”

(4) Many of the quantitative aspects of foundational measures used to interpret PAWS were lacking, for example, how non-overlapping response is defined in the LDA and statistical analysis for interpretations of significance. Please also see the full Comments from the Reviewers, included below. We do appreciate that you could make revisions that would address some of the reviewers' specific comments, but our determination is that addressing these concerns would be beyond the scope of a revision at eLife.

We agree: it was unclear, both because the ellipses on the original figures were overlapping, and because the two-dimensional (LD1/LD2) analysis was confusing. We have completely reworked the PAWS analysis pipelines using a univariate measure instead.

Using the revised, univariate measure we report the discriminatory power to distinguish between pain (HP and LP treatments) and no-pain (DB and CS) treatments, which is 85% on average. Additionally, we report Leave-One-Out (LOO) cross-validation of pain / nonpain discriminatory power, which was 84%, with [79%,88%] nonparametric bootstrap 95% CI. And we also separately report the power to determine whether the stimuli treatment was no-pain (DB and CS) versus low pain (LP) versus high pain (HP), which is 69% on average (LOO 66%, with [60%,72%] 95% CI).

The revised figures show the clear separation between painful and non-painful stimuli along a unidimensional axis.

Although the revised main text focuses on the binary discrimination task between pain and no-pain stimuli, the same univariate pain scale be used further to discriminate between light pain (light pinprick stimulus) and heavy pain (heavy pinprick treatment). To demonstrate this, we include a supplementary figure that shows the accuracy of this trinary discrimination task.

Reviewer #1:This work described in the manuscript by Jones, Foster, et al., describes an automated protocol to quantify and interpret (based on 8 components of the paw trajectory: 4 pre and 4 post) the response of mice to noxious and innocuous mechanical stimuli applied to the plantar hind paw using high speed video recordings. The protocol is designed to objectively report whether the mouse interprets the stimuli as non-painful or painful as well as the severity or intensity of the pain. Mouse strains that show extremes in their response to the stimuli (hypo and hyper) and also eDREADD activation of the pain neuronal ensemble within the amygdala were used to validate whether the protocol is able to accurately quantify and interpret a range of stimuli.Comments:The development of an easily accessible, automated and objective way to measure evoked mechanical pain in rodents will be extremely valuable for the field, as will delineating affective versus discriminative components (and in the context of chronic pain) and this is a good first step.1) Authors should discuss variability in PAWs in relation to the variability inherent in the methods used to simulate the paw.

This point is now added in the Discussion section.

2) PAWS data in Figure 4 are showing significant differences between CS+DB and LP+HP in the scoring but the LD1/LD2 analyses extract differences in the LP and HP for post-peak features. A deeper explanation of how data are to be reported be interpreted would be helpful.

We have completely re-worked the PAWS analysis and software to produce a simple, univariate measure of pain, derived from the features extracted from high-speed video. The paper now directly addresses the ability of this scale to detect pain vs no-pain, and also, separately, high pain (HP) vs low pain (LP).

3) Activation of the pain ensemble with the DREADD and CNO evokes a pain like behavior. I would have expected that the LP would have resulted in greater shaking and guarding behavior than the -CNO control. Do the authors have an explanation?

This is an astute observation made by the reviewer and we have now added this explanation in the Discussion section:

“Based on our single-neuron microendoscope calcium imaging data in the basolateral amygdala (Corder, 2019), the nociceptive neurons respond only to stronglynoxious sensory stimuli. Only after a peripheral nerve damage to induce chronic neuropathic pain, did we observe that formerly-innocuous stimuli engage this BLA ensemble. This suggests that some up-circuit plasticity occurred that re-directed touch information into the BLA ensemble (e.g. opening of the spinal “gate”, see Torsney and MacDermott, 2006). Thus, without this central plasticity the LP stimulus likely does not activate the ensemble under normal, uninjured conditions. Therefore, chemogenetic activation of the nociceptive ensemble is not necessarily predicted to amplify or modulate neural processes related to non-noxious stimuli. We did observe some spontaneous lifting and licking of the hindpaws upon CNO treatment, which could reflect general aversive perception, but how this might alter withdrawal reflexes that use distinct neural circuits is unclear. The fact that we observed a robust increase in paw shaking and guarding-related behaviors supports the conclusion that the BLA nociceptive ensemble is specifically tuned to encode and modulate nociception only.”

5) The 0.7 probability correct shown in Figure 5 and Figure 6 as being sufficient should be interpreted for the reader a bit more than what is provided in subsection “Automated scoring of rapid paw dynamics and lingering pain behaviors”. Related to this: There are no stats for these comparisons.

We have updated the text to clarify these reported results. In the revised version of PAWS, based on a univariate pain scale, we can discriminate no-pain vs pain treatments with 85% accuracy, in a model fitted to all the data.

Later in the manuscript, we also report the “leave-one-out cross-validated” accuracy, which involves omitting each mouse (or an entire strain) when fitting the model, and then attempting to predict the pain treatment of the mouse (or strain) that had been left out. In cross validation our accuracy is slightly reduced – to about 84% (when leaving out a single mouse) or 81% (when leaving out an entire strain), with nonparametric bootstrap 95% confidence intervals of [79%,88%] and [77%,86%], respectively. These results reflect the accuracy that researchers can expect to achieve when scoring pain treatments of mice (or strains) not included in the model fitting.

6) The rationale for mentioning a potential relationship to aggressive behavior in the Discussion section is not well-developed. What about a relationship of PAWS outcomes to the magnitude of a startle reflex?

This is good point raised by the reviewer and we have updated the discussion to further develop the aggression linkage, and we have included the reviewer’s idea about potential startle reflexes in the SJL strain driving increased measurements with PAWS.

Reviewer #2:In this article, Jones et al. describe an automated assessment of tactile sensitivity. They report development and validation of a novel combination of high-speed videography and automated paw tracking. With this resource, they demonstrate that paw withdrawal to innocuous versus noxious stimuli can be separated in six inbred mouse strains. Using this system, they also present evidence that activating an ensemble of basolateral amygdala neurons during noxious stimulation changes paw withdrawal metrics. This approach represents a significant advance in behavioral assessment of tactile sensitivity, with the potential to contribute to much-needed discoveries in this domain.Major comments:2) It would benefit the readers to state more clearly that analgesic reversal of pain-related changes to paw withdrawal were detected in Abdus-Saboor et al., (2019), wherein a software-assisted, manual scoring system of similar indices was used. Reversal of changes in paw withdrawal-associated behaviors is a robust indicator that they are pain-related.

It is not clear to us what the reviewer is requesting here in referencing Abdus-Saboor et al., (2019). In that context, we used a CFA model and activated polymodal nociceptors, and while giving a cocktail of an opioid and anti-inflammatory agent, we observed that this analgesic cocktail reduced optogenetically induced pain behaviors. We don’t use a similar model in this work, and thus the instructions here are not clear. However, based on Abdus-Saboor et al., (2019)and work from many colleagues in the field, we believe it is clear that the features that we automatically measure with PAWS (paw shaking, paw guarding, paw height, paw velocity) are indeed pain-related.

5) In subsection “Statistical modeling with linear discriminant analyses separates touch versus pain across six inbred mouse strains” the authors note that pre-peak paw movements are nocifensive in nature, whereas those behaviors seen post-peak (shaking/guarding) are supraspinal in origin. I agree that this distinction seems plausible, given the timescales involved, but are there data in the literature to support this? Unless this is founded upon findings from prior studies, this comment might be more at home in the Discussion section.

This is a good point made by the reviewer and we have moved this to the Discussion section instead of the results.

Reviewer #3:Without such a measure, the current implementation of PAWS is of limited general use for those who are interested in studying pain in mice. For example, what endpoint(s) is someone in the field supposed to use when studying pain and responses to analgesics? Y-velocity? X-velocity? # paw shakes? Paw height? All of these variables? Two of these variables? Three? Four? It is confusing and not simplified.

Yes, we agree: it was confusing. In the revised work there is a single, simple endpoint: a univariate pain scale, which is automatically computed for each mouse.

As we show in the revision, this univariate pain scale allows for accurate scoring of painful versus non-painful stimuli (85% accuracy).

It is still true that the computation of this univariate pain scale combines several behavioral features like x-velocity and #paw shakes, etc – but these features are automatically extracted by the software and the resulting endpoint is a simple, univariate pain scale.

The title of the paper implies the machine vision approach is automated; however, this does not appear to be the case. The first step in this process is to manually label the center of the stimulated paw. This is thus more akin to a "semi-automated" approach.

We respectfully disagree with the reviewer about the manuscript’s title, as the PAWS software automatically measures a pain scale using machine-vision approaches to track the paw. All such machine-vision approaches require some initial input to indicate the object being visualized, but there is no manual annotation of features or manual scoring of pain.

Time of the first withdrawal peak (t*) is a critical variable in their analytic pipeline. However, it is unclear precisely how t* is calculated or defined. In the paper, authors write that t* is the time leading up to the initial paw peak. And in Figure 3A, since this is a 3d graph in 2d, it is unclear where t* is relative to the trajectory data.

Thank you for asking us to clarify. The revised manuscript defines *t** clearly: it is the time of the first peak in paw height (y-axis). All times following *t** are called “post-peak”, and all times prior to *t** are called “pre-peak”.

We agree that time *t** is difficult to visualize directly, but the definition itself is clear (and it is also codified concretely in the software code that accompanies the paper, which will be made freely available to all researchers).

From the LDA analyses, the authors state that the low and high pain stimuli separate from the non-pain stimuli (Figure 5A,B). However, I do not see a clear separation between these groups in the figure. Instead there appears to be significant overlap, which raises the question as to how specific the LDA analysis is at discriminating, in a quantitative manner, the magnitude of a pain response.Figure 1B. The paw withdrawal frequency data has no error bars and the number of mice used to generate these data is not indicated. The authors write that strains differ in some of these assays but provide no statistics to confirm that the differences shown are statistically significant.

Thank you, the revised work includes N numbers of mice used in the figure and figure legend and a direct statistical test that shows a significant difference in the accuracy of determining pain stimulus in the SJL strain versus other strains (significant difference in mean classification accuracy between SJL and DBA1; one-sided Welch two sample t-test t=2.9406, df=36.894, p=0.003).

Moreover, the authors are encouraged to consult and cite work by Jeff Mogil's group who evaluated mechanical sensitivity in different mouse strains many years ago.

We have consulted and cited Jeff Mogil’s work in relation to genetic strain differences (see Discussion section).

Figure 3B and C are difficult to interpret. Provide more details in legend and in the figure itself. Ex. in B, are those lines with two arrow heads? What does the length of the line and angle of the line mean? And in C, how was the shaking vs guarding bout determined? Was this done by a human or did the algorithm make these assessments in an unbiased manner?

Periods of guarding and shaking (and also the number of shakes) are determined automatically by the software. Indeed, the entire PAWS platform is automatic with no human input required.

We have clarified the definition of shaking vs guarding periods in the caption to Figure 3. But the basic idea is simple: shaking is a series of back-and-forth movements of the paw in the principle direction of movement, whose distance exceeds 35% of the first paw peak height; and all other periods of time outside of these shakes are scored as guarding.

Yes, the arrows in Figure 3B have arrowheads on both sides: they represent the vector (in x-y plane) of principle movement for that window of time. The arrows have constant length. We have clarified caption.

Figure 6. Authors state 129 and SJL mice are outliers, but based on data presented in this figure, it is hard to appreciate how exactly they are outliers. In panel B, 129 mice show a similar probability correct relative to all other strains, and SJL error bounds largely overlap the other strains. An outlier is typically defined as being two or more standard deviations from the mean.

You are absolutely correct: strain 129 was not in fact a statistical outlier, and so we have removed any such claims or discussion from the manuscript altogether. Strain SJL is indeed an outlier (it falls outside the 95% CI for other strains, and a t-test comparing to strain DBA1 is significant; one-sided Welch two sample t-test t=2.9406, df=36.894, p=0.003).

Moreover, the atypical withdrawal response may have nothing to do with a pain hyposensitivity phenotype, as the authors assert. Instead, these strains may simply have motor deficits that prevent them from performing more vigorous/elaborate paw withdraw responses.

We agree, and thank for pointing this out. It is possible that abnormal pain responses result from primary differences outside of the pain neuroaxis. Further studies will need to be performed to determine if this is the case, as described in the revised manuscript.